# Urease of *Aspergillus fumigatus* Is Required for Survival in Macrophages and Virulence

Zhenzhen Xiong,[a] Nan Zhang,[a] Liru Xu,[a] Zhiduo Deng,[a] Jarukitt Limwachiranon,[a] Yaojie Guo,[a] Yi Han,[a] Wei Yang,[d] Daniel H. Scharf[a,b,c]

[a]Department of Microbiology, School of Basic Medical Sciences, Zhejiang University School of Medicine, Hangzhou, China
[b]The Children's Hospital, Zhejiang University School of Medicine, National Clinical Research Center for Child Health, Hangzhou, China
[c]Key Laboratory of Immunity and Inflammatory Diseases of Zhejiang Province, Hangzhou, China
[d]Department of Biophysics and Department of Neurosurgery, The Fourth Affiliated Hospital, Zhejiang University School of Medicine, Hangzhou, China

Zhenzhen Xiong and Nan Zhang contributed equally. Author order was determined by alphabetically.

**ABSTRACT** The number of patients suffering from fungal diseases has constantly increased during the last decade. Among the fungal pathogens, the airborne filamentous fungus *Aspergillus fumigatus* can cause chronic and fatal invasive mold infections. So far, only three major classes of drugs (polyenes, azoles, and echinocandins) are available for the treatment of life-threatening fungal infections, and all present pharmacological drawbacks (e.g., low solubility or toxicity). Meanwhile, clinical antifungal-resistant isolates are continuously emerging. Therefore, there is a high demand for novel antifungal drugs, preferentially those that act on new targets. We studied urease and the accessory proteins in *A. fumigatus* to determine their biochemical roles and their influence on virulence. Urease is crucial for the growth on urea as the sole nitrogen source, and the transcript and protein levels are elevated on urea media. The urease deficient mutant displays attenuated virulence, and its spores are more susceptible to macrophage-mediated killing. We demonstrated that this observation is associated with an inability to prevent the acidification of the phagosome. Furthermore, we could show that a nickel-chelator inhibits growth on urea. The nickel chelator is also able to reverse the effects of urease on macrophage killing and phagosome acidification, thereby reducing virulence in systemic and trachea infection models.

**IMPORTANCE** The development of antifungal drugs is an urgent task, but it has proven to be difficult due to many similarities between fungal and animal cells. Here, we characterized the urease system in *A. fumigatus*, which depends on nickel for activity. Notably, nickel is not a crucial element for humans. Therefore, we went further to explore the role of nickel-dependent urease in host-pathogen interactions. We were able to show that urease is important in preventing the acidification of the phagosome and therefore reduces the killing of conidia by macrophages. Furthermore, the deletion of urease shows reduced virulence in murine infection models. Taken together, we identified urease as an essential virulence factor of *A. fumigatus*. We were able to show that the application of the nickel-chelator dimethylglyoxime is effective in both *in vitro* and *in vivo* infection models. This suggests that nickel chelators or urease inhibitors are potential candidates for the development of novel antifungal drugs.

**KEYWORDS** antifungals, *Aspergillus fumigatus*, urease, virulence, nitrogen metabolism

Address correspondence to Daniel H. Scharf, dhscharf@zju.edu.cn.

The authors declare no conflict of interest.

[This article was published on 14 March 2023 with an error in the article text. The text was corrected in the current version, posted on 27 March 2023.]

Fungal diseases are still a severe public health problem worldwide, and invasive aspergillosis (IA), which is mainly caused by the opportunistic mold *Aspergillus fumigatus*, is one of the most common life-threatening fungal diseases in immunocompromised patients (1, 2). Despite great efforts in prophylactic strategies, diagnostic tests,

and antifungal treatments, the morbidity of invasive aspergillosis remains high, with the mortality rate ranging between 30% and 95% (3–6). In recent years, many virulence factors of *A. fumigatus* have been identified, and many of these are required to either interact with the host immune system or adapt to environmental stresses in the host (7). Additionally, an increasing number of genes that are involved in fungal nutrition (e.g., nitrogen metabolism-related genes) have been associated with *A. fumigatus* virulence (8).

The enzyme urease catalyzes the hydrolyzation of urea into ammonia and bicarbonate. This reaction is of great importance for agriculture because it is the only means by which plants are capable of using nitrogen from urea (9). On the other hand, microbial ureases can cause large amounts of ammonia emission from urea-based fertilizers, which results in a loss of valuable nitrogen and negatively affects the atmosphere and the surrounding environment (10). The urease from jack bean (*Canavalia ensiformis*) was the first enzyme to be crystallized and shown to contain nickel (11, 12). However, thus far, the best data on urease structure and maturation were obtained from bacteria, in particular, *Klebsiella aerogenes* and *Helicobacter pylori* (13). Typically, the minimal functional structures of bacterial ureases are comprised of two or three subunits, whereas in plants, all of the subunits are fused into one polypeptide chain (14). Despite the difference in subunit composition, all ureases possess similar tertiary structures, and the urease active site, where two nickel ions are coordinated by a carbamylated lysine, four histidine residues, and one aspartate residue, is highly conserved (14). Urease is initially synthesized as an inactive apoenzyme, to become enzymatically active, it must undergo a maturation process that involves carbamylation of the active-site lysine followed by delivery of two nickel ions into the active site (13).

The maturation of urease in bacteria is carried out by four accessory proteins, namely, UreD (called UreH in *H. pylori*), UreE, UreF, and UreG. UreE is known to be a dimeric nickel-binding protein that supplies nickel to urease during the maturation process (15, 16). UreG is a chaperone and a SIMIBI (signal recognition particle, MinD, and BioD) class GTPase that is responsible for the GTP (guanosine-5′-triphosphate) hydrolysis that is associated with the transfer of $CO_2$ to the active-site lysine (17). UreF appears to gate the GTPase activity of UreG to enhance the fidelity of the urease activation (18). It has been proposed that the nickel-bound UreE dimer can bind two UreG monomers in the presence of GTP and $Mg^{2+}$ to form the $UreE_2G_2$ complex and thereby trigger nickel translocation from UreE to UreG (19). UreF forms a $UreF_2D_2$ complex with UreD in a 2:2 stoichiometry, and the $UreF_2D_2$ complex competes with $UreE_2$ for nickel-charged $UreG_2$ to form the $UreG_2F_2D_2$ complex, which further forms a $UreG_2F_2D_2$-apourease activation complex through a direct interaction between UreD and apourease, after which GTP hydrolysis by UreG is catalyzed to complete the final step of the insertion of nickel into the apourease (17). This transfer between the nickel binding site of the UreG dimer and the urease is mediated by a transfer tunnel in the $UreF_2/D_2$ complex, and amino acid mutations in the tunnel can greatly reduce the urease activity (20–23). In *H. pylori*, UreE can receive its nickel from the hydrogenase maturation factor HypA, suggesting that crosstalk exists between the urease and [NiFe]-hydrogenase maturation pathways (19, 24, 25). Homologs of UreD, UreF, and UreG have also been identified in plants and have been shown to be essential for urease maturation after a stepwise assembly (26, 27). Interestingly, a UreE homolog was not found in plants, and the plant UreG with HXH motifs was postulated to combine the functions of bacterial UreE and UreG (28). It was reported decades ago that the utilization of urea by fungal cells is entirely urease-dependent (29). However, fungal urease systems have not been noticed for a long time; however, recently, the factors required for urease activation in *Cryptococcus neoformans*, which is a yeast pathogen that causes meningoencephalitis, were described (30).

Urease has been shown to function as a general virulence factor for many pathogens. Since urea occurs in all body fluid compartments (31, 32), urease often plays a role in the nitrogen assimilation of pathogenic microbes under nutrient-limited conditions (33, 34). Beyond this, urease activity can cause a local environmental pH increase

to ensure the colonization and survival of pathogens. As a typical example, the bacterial pathogen *H. pylori* survives in the acidic gastric mucosa via the hydrolysis of urea neutralizing the proton inflow into the periplasm (35, 36). In *Proteus mirabilis*, the products that are liberated from urea hydrolysis form urinary stones with magnesium and calcium ions, which in turn provide protection for the pathogen (37). Recently, urease was also found to be required not only for the acid response network of *Staphylococcus aureus* but also for a persistent murine kidney infection (38). Studies on the role of urease for pathogenic fungi suggest that, in addition to urease-dependent alkalization, ammonia toxicity to host cells can worsen fungal infections. During the parasitic cycle of *Coccidioides posadasii*, the ammonia produced by active urease from the spherules results in localized lung tissue damage and the exacerbation of the respiratory disease in mice (39, 40). Consistently, ammonia-induced host tissue damage also occurs in infections by bacteria (41). Moreover, the toxic effect of ammonia on human epithelial/endothelial cells may promote systemic disease. This was proposed as the underlying mechanism for the finding that urease promotes the transmigration of *C. neoformans* across the blood-brain barrier, thereby enhancing the invasion of the central nervous system (30, 42, 43).

*A. fumigatus* is a leading cause of mortality in immunocompromised patients. Over 95% of the clinical isolates are urease-positive (44, 45). However, the biochemical properties and the role of urease in the virulence of *A. fumigatus* remain unknown. Here, we validated urease as a potential druggable target by demonstrating a critical role for this enzyme in growth on urea as a nitrogen source, immune evasion, and virulence. We further demonstrated the suitability of this target by using a nickel chelator in both *in vitro* and *in vivo* infection models.

## RESULTS

**Identification of the genes encoding urease and maturation proteins in *A. fumigatus*.** In order to investigate the function of urease and the maturation pathway, we first identified the involved genes from the genome of *A. fumigatus*. Based on the annotated gene information in the FungiDB database, we found the urease gene *ureB* (AFUA_1G04560) and the maturation genes *ureD* (AFUA_2G16070) and *ureG* (AFUA_2G12900). The third gene, namely, *ureF*, that is needed for urease maturation in other species, is missing here. Nevertheless, we identified an ortholog of *ureF* in the *A. fumigatus* genome via the tblastn program by using *Arabidopsis thaliana* UreF as a query (30% amino acid identity, 50% amino acid similarity) (Fig. S1), and we designated this new gene AFUA_6G04385.

To gain a better understanding of the phylogenetic relationship of *A. fumigatus* UreB, a phylogram was constructed using homologous amino acid sequences from other fungi, plants, and the $\alpha$ subunit of bacterial ureases (Table S3). Three distinct clades of urease sequences, corresponding to the bacterial, plant, or fungal origin were identified (Fig. S2). An alignment of these proteins showed the presence of one lysine residue, four histidine residues, and one aspartate residue in *A. fumigatus* UreB, which are crucial for the catalytic activity of bacterial and plant ureases (Fig. S3). Two nearby histidine residues that probably assist in substrate binding (46) are also conserved in *A. fumigatus*.

We further performed sequence alignment for the urease maturation proteins. *A. fumigatus* UreD shares a low level of sequence similarity with its homologs (Fig. S4), but it has conserved residues that have been genetically and biochemically characterized to be essential for urease activation in bacteria and have been proposed to be involved in the internal transfer of nickel (21–23). Sequence comparisons of UreF homologs revealed the strongest conservation in the C-terminal part and in the region near the N terminus (Fig. S5), which encompass the residues constituting the UreG binding site (18, 47). UreG appears to be the most highly conserved accessory protein. All of the UreG proteins that were analyzed here contain the regions that are responsible for the binding of GTP, the binding of nickel, and the interaction with UreF (Fig. S6). In accordance with other eukaryotes, *A. fumigatus* lacks a homolog of UreE that is necessary for bacterial urease maturation (28, 30).

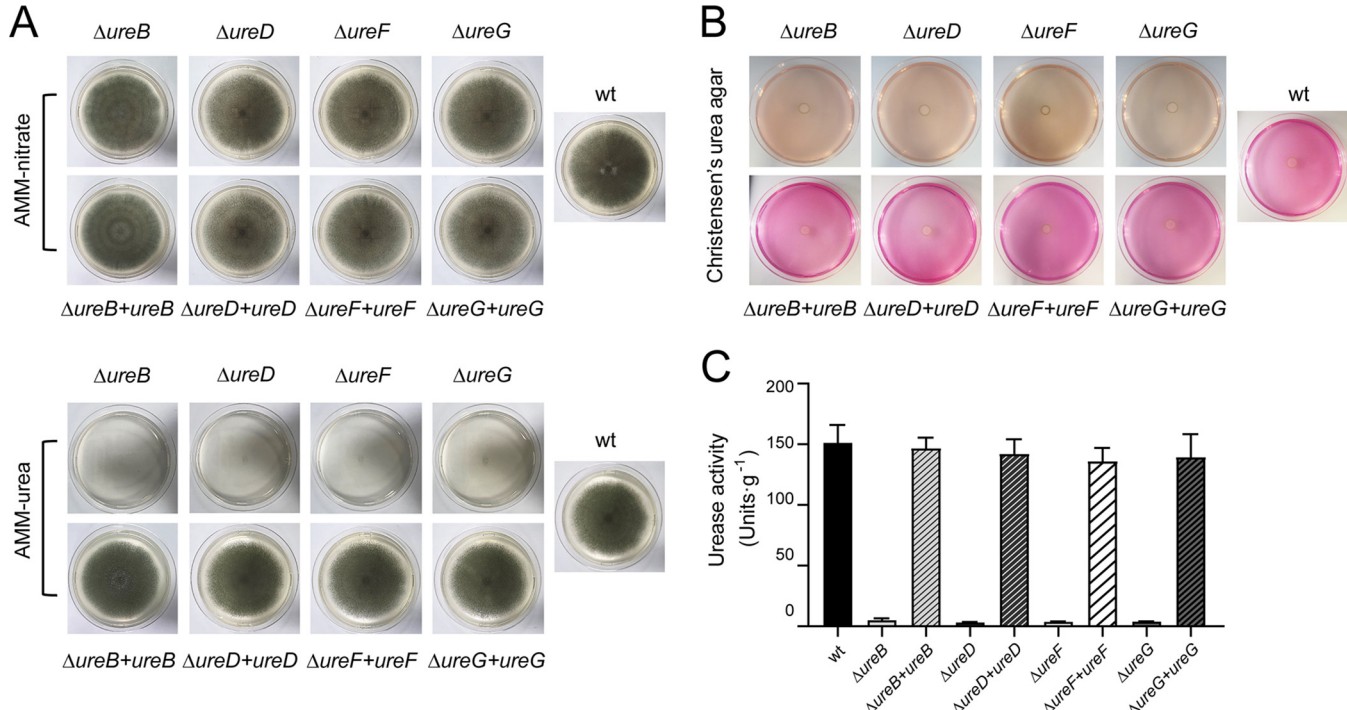

**FIG 1** Inactivation of urease structural and maturation genes abolishes urease activity in *A. fumigatus*. (A) Growth tests of the wild-type as well as the *ureB*, *ureD*, *ureF*, and *ureG* mutant and complemented strains on AMM agar supplemented with nitrate or urea as the sole nitrogen source. Photographs were taken after 7 days of inoculation with $5 \times 10^3$ spores on each plate. (B) Urease activity tests of all mutant and complemented strains on Christensen's urea agar. Production of active urease induces a pH shift of the medium, which is visualized by a pH-responsive dye. (C) Urease activity of cell extracts from the wild-type, urease maturation gene deletions, and corresponding complemented strains. The results that are shown are the averages of assays performed in triplicate. Error bars denote standard deviations.

**UreB and the maturation proteins UreD, UreF, and UreG are required for normal growth on urea as the sole nitrogen source.** It was known that an active urease is crucial for the growth of urease-producing *Aspergillus* on urea as the sole nitrogen source (48). To confirm this, we generated a *ureB* null (named Δ*ureB*) and deletions of the maturation genes (Δ*ureD*, Δ*ureF*, Δ*ureG*) in *A. fumigatus* CEA17Δ*akuB*^KU80 (49, 50). The corresponding complemented strains were subsequently achieved via the reintroduction of each gene (Fig. S8–S11). When grown on *Aspergillus* Minimal Medium (AMM) with nitrate as nitrogen source, all of the mutants, as well as the complemented strains, exhibited a similar growth phenotype in comparison to the wild-type. However, when grown on AMM with urea as the nitrogen source, a normal growth was observed for the wild-type and complemented strains but not for the Δ*ureB*, Δ*ureD*, Δ*ureF*, and Δ*ureG* mutants (Fig. 1A; Fig. S12). In addition, the dry weight is the same for the wild-type, Δ*ureB*, and complemented strains grown in liquid AMM nitrate (Fig. S16). The cultivation of all, of the strains on Christensen's urea agar, showed that a pink colorization of the medium due to a urease-induced pH increase was only caused by the wild-type and the reconstituted isolates (Fig. 1B). We further measured the urease activity of the raw extract from the mycelium of each strain using a colorimetric assay, based on the production of ammonia. None of the deletion strains showed urease activity, whereas the complemented strains recovered enzyme activity that was comparable to that of the wild-type (Fig. 1C). These results suggest that UreB and the accessory proteins UreD, UreF, and UreG are indispensable for the use of urea as the sole nitrogen source.

**UreB is a cytoplasmatic protein, and its transcript and protein levels are upregulated by urea.** The *ureB* gene encodes the urease structural protein, and its expression is likely to be responsive to the availability of urea in the environment. To investigate the regulation of *ureB*, we assayed the relative gene expression level after growth in liquid AMM with nitrate, urea, or ammonium. As shown in Fig. 2A, the transcript level of *ureB* increased 3.5-fold with urea as the nitrogen source, compared to nitrate or ammonium

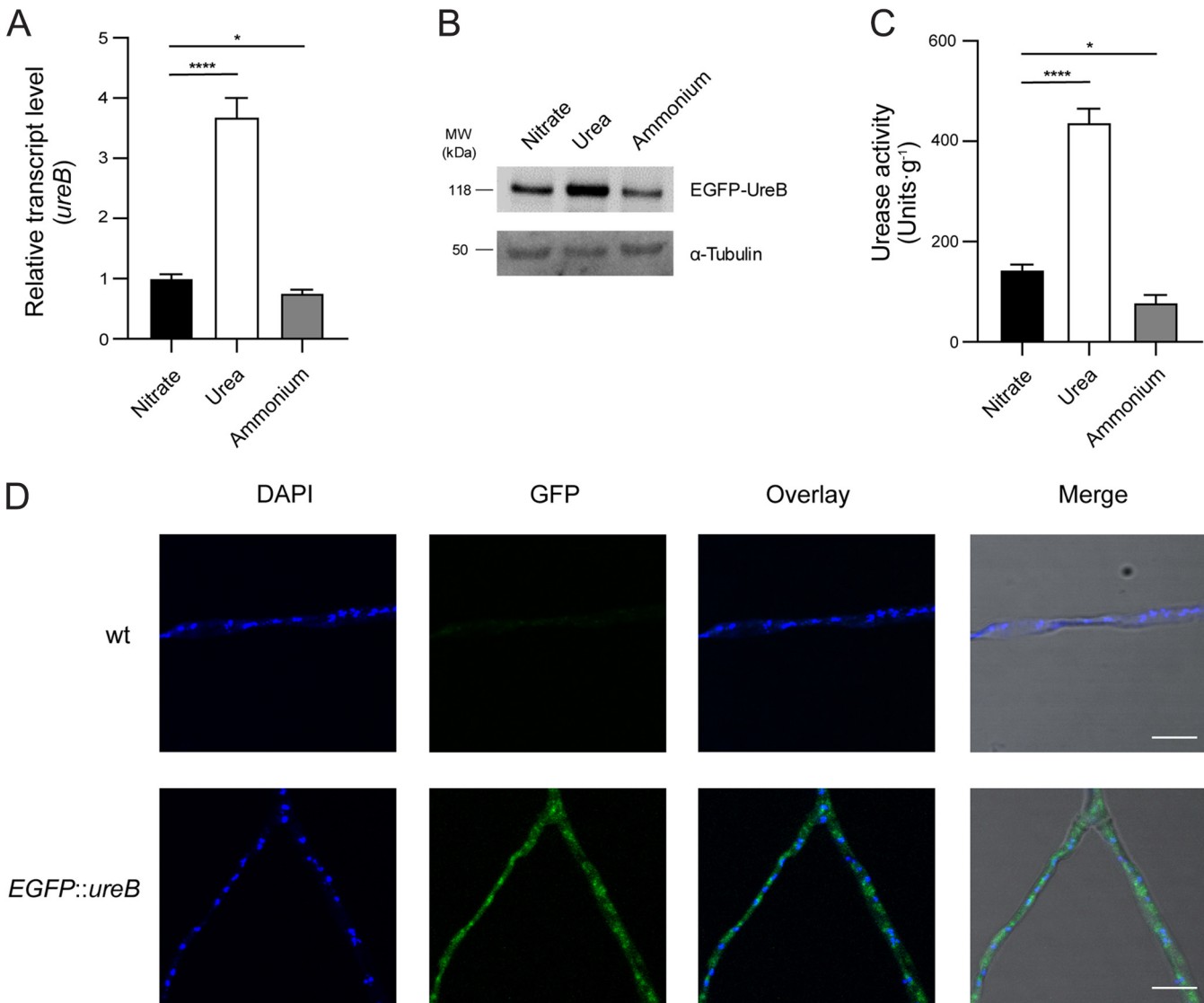

**FIG 2** The urease activity of *A. fumigatus* increases with urea as the sole nitrogen source, in comparison to nitrate and ammonium. (A and B) The relative expression of *ureB* was determined via real-time RT-PCR (panel A), and the protein level was determined via Western blot (panel B). Total RNA/protein were extracted from wild-type or *EGFP::ureB* strains grown in AMM with nitrate, urea, or ammonium for 24 h. The $\beta$-actin gene was used as an internal control to normalize the transcript level of *ureB* that was expressed under different conditions. A western blot analysis was performed using anti-GFP and anti-tubulin antibodies. (C) Urease activity of *A. fumigatus* wild-type with nitrate, urea, or ammonium as the nitrogen source. The results represent the mean $\pm$ SD (standard deviation) from three independent biological replicates. Statistical significance was assessed using a one-way ANOVA test with multiple comparisons. *, $P < 0.05$; ****, $P < 0.0001$. (D) The fluorescent signals of EGFP-UreB and DAPI were observed via confocal microscopy after growing the strains with urea for 24 h. Scale bar, 10 $\mu$m.

as the nitrogen source. For a verification at the protein level, we constructed an N-terminal fusion of UreB with the enhanced green fluorescent protein (EGFP) (Fig. S13A–D). A Western blot analysis using anti-GFP antibody indicated an increase in the amount of EGFP-UreB in mycelium grown in AMM urea (Fig. 2B). As a consequence of the increase in UreB expression, the urease activity of the strain cultured in AMM urea was significantly higher than that in AMM nitrate/ammonium (Fig. 2C). Using confocal microscopy, we were able to show that EGFP-UreB is localized in the cytoplasm of *A. fumigatus* (Fig. 2D). This is in agreement with previous observations in other urease-containing organisms (35, 39, 51). Intriguingly, the fluorescent signal of EGFP-UreB was also observed in resting conidia (Fig. S13E). Moreover, conidia harvested on AMM agar with urea as the nitrogen source exhibited the highest urease activity (Fig. S13F), compared to nitrate or ammonium.

**Interaction of UreB and its maturation proteins in *A. fumigatus*.** The maturation pathway of urease consists of a series of interacting proteins that lead to the delivery and incorporation of nickel into the active site of the enzyme. In an attempt to clarify the interaction between *A. fumigatus* urease and its maturation proteins, we constructed N-terminal tandem affinity purification (TAP)-tagged versions of UreB (nTAP-UreB) and UreF (nTAP-UreF) as well as C-terminal TAP-tagged versions of UreD (UreD-cTAP) and UreG (UreG-cTAP) (Fig. S14 and S15). All of the TAP-tagged proteins could be successfully purified from protein extracts, as shown by a Western blot using the anti-calmodulin binding protein epitope tag antibody (Fig. 3A). The silver staining of the proteins that were copurified with nTAP-UreB revealed the presence of nTAP-UreB, UreD, and UreF. The silver-stained gel also clearly illustrated the presence of UreB, UreD-cTAP/UreD, and UreF/nTAP-UreF in the UreD-cTAP or nTAP-UreF purified fractions. However, none of the TAP purifications isolated the UreG protein. Meanwhile, the fraction purified from the *ureG::cTap* strain showed UreG-cTAP alone in a small amount (Fig. 3A). All of the protein lanes were excised from silver-stained gel and submitted for mass spectrometry-based identification. A MS analysis confirmed the associations between UreB, UreD, and UreF, regardless of which one was labeled by the TAP tag, whereas, apart from UreG, no UreB or other accessory protein could be identified in the fraction from *ureG::cTap* (Fig. 3B; supplementary MS table). Neither urease nor the accessory proteins could be purified from wild-type mycelium or from a strain with a TAP-tagged subunit of the NOT complex, which was used as a control. We continued to test whether the final eluted fractions had ureolytic activity. As Fig. 3C shows, eluates of nTAP-UreB, UreD-cTAP, and nTAP-UreF demonstrated marked enzyme activities at different levels, but the UreG-cTAP enriched fraction did not present any activity. Taken together, our results indicate that UreD and UreF can form a stable complex with UreB, and UreG may transiently bind to this complex to facilitate the activation of urease.

**UreD, UreF, and UreG are necessary for the activation of *A. fumigatus* urease in *Escherichia coli*.** To further establish the importance of *A. fumigatus* accessory proteins in urease activation, we attempted to reconstitute the urease activity in a heterologous host. For this purpose, cDNA sequences coding for UreB, UreD, UreF, and UreG were cloned and fused with *Strep*, *MBP*, *His*, and *HA* tags, respectively. Strep-tagged UreB was then coexpressed with all three or with any two of the recombinant accessory proteins $UreD_{MBP}$, $UreF_{His}$, and $UreG_{HA}$ in *E. coli* BL21(DE3), which is not capable of producing urease (26). After Strep affinity purification, $UreB_{Strep}$ and its associated proteins were visualized in Western blots with antibodies against the respective tags (Fig. 4A). The copurification of UreF and/or UreG with UreB relies on the presence of UreD, suggesting that UreD may directly interact with UreB and serve as a platform for the other two proteins binding to UreB. A subsequent urease activity measurement was performed for all of the purified fractions. As indicated in Fig. 4B, enzyme activity was only detected in the fraction containing UreB, UreD, UreF, and UreG. Consistent with this result was the observation that the *E. coli* strain simultaneously expressing UreB and the three accessory proteins could induce a pink colorization on Christensen's urea agar, whereas the other strains could not (Fig. 4C). Our data suggest that the accessory proteins UreD, UreF, and UreG of *A. fumigatus* are not only necessary but also sufficient for urease activation.

**Urease is important for *A. fumigatus* virulence.** Given the important metabolic role of urease and the high proportion of urease-positive *A. fumigatus* clinical isolates, we hypothesized that urease might contribute to the virulence of *A. fumigatus*. To test this hypothesis, we assessed the virulence of the Δ*ureB* strain in mice that were treated with cortisone acetate and retained the ability to recruit neutrophils and monocytes. Out of all of the mice that were infected with the wild-type strain in a lung infection model, 100% succumbed to the disease within 4 days after infection. In contrast, 50% survival was seen in the cohort that was infected with the Δ*ureB* mutant. Infection with the reconstituted strain resulted in complete killing at 3 days postinfection (Fig. 5A). Out of all of the mice that were infected with the wild-type strain in a systemic infection model, 33% survived, and the rest succumbed to the disease. In contrast, 73% survival was seen in the cohort infected with the Δ*ureB* mutant. Infection with the

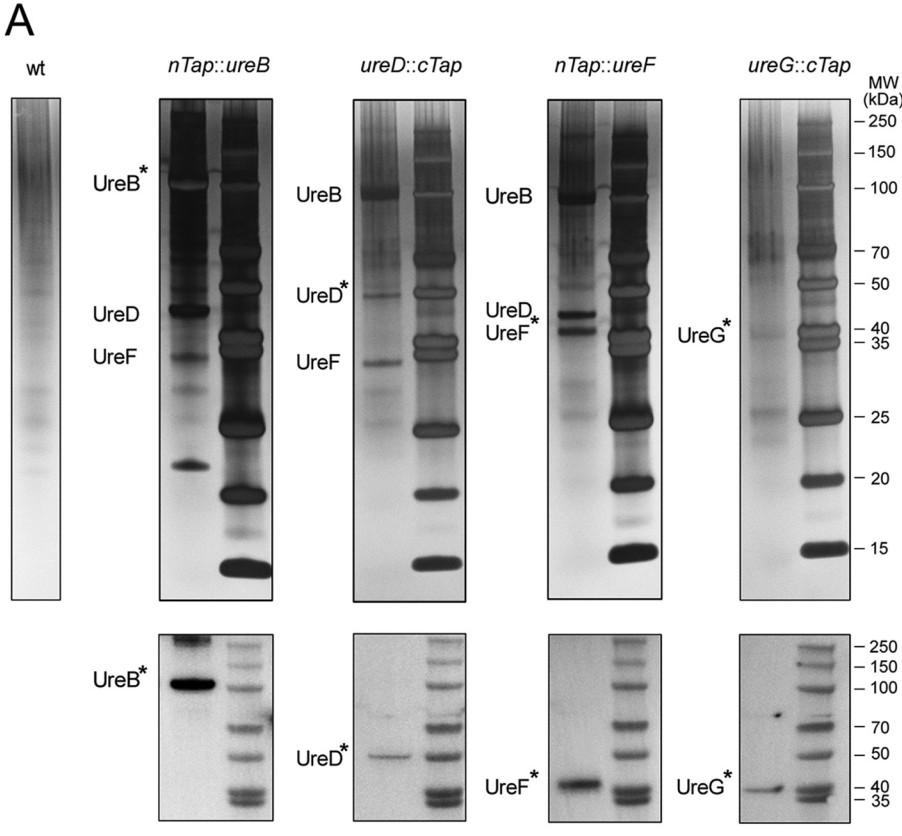

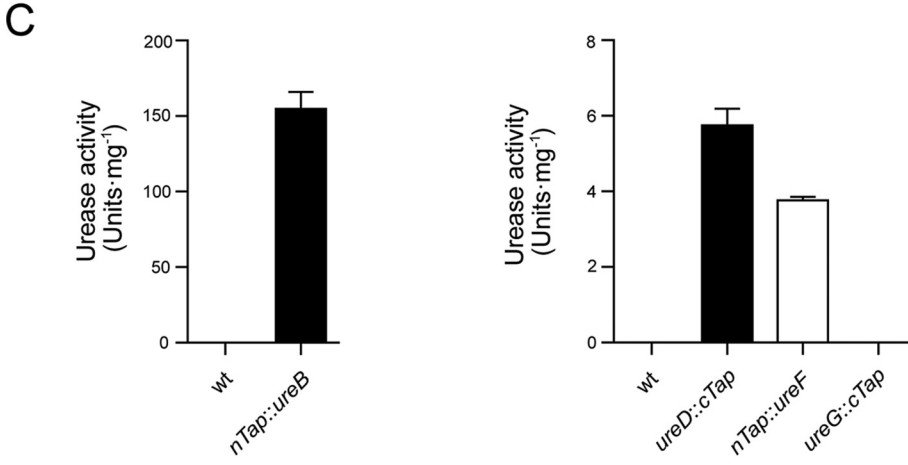

**FIG 3** The accessary proteins UreD and UreF interact with UreB in *A. fumigatus*. (A) TAP purified fractions were analyzed by 4% to 12% SDS-polyacrylamide gel electrophoresis with silver staining (top) and were

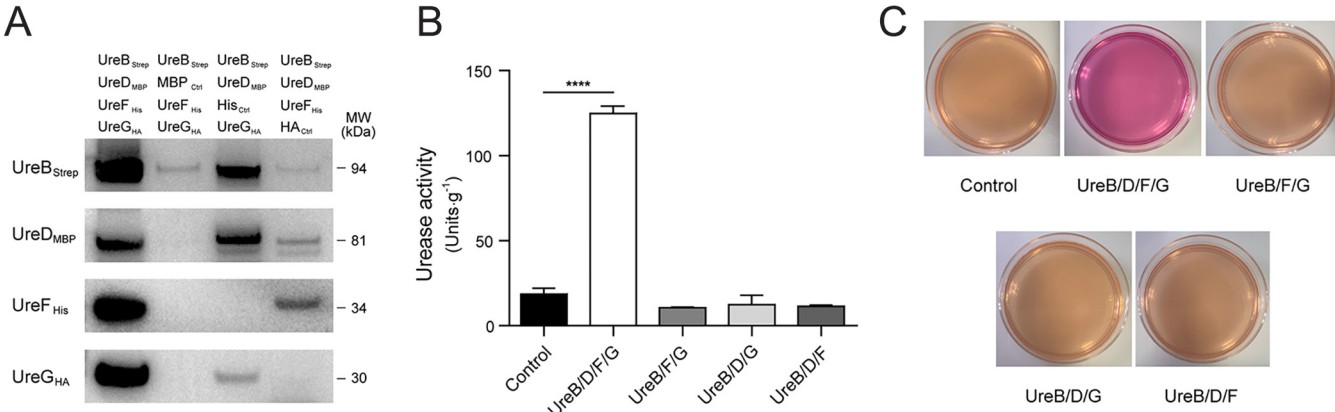

**FIG 4** The coexpression of UreD-UreF-UreG facilitates the heterologous activation of UreB in *E. coli*. (A) Strep pulldown results were analyzed via Western blot. Strep-tagged UreB was coexpressed with all three or any two of the recombinant accessory proteins, namely, UreD$_{MBP}$, UreF$_{His}$, and UreG$_{HA}$, as specified in the legend. Antibodies against Strep, MBP, His, and HA tags were used. (B) The enzymatic activity of the coeluted proteins was measured. The results represent the mean $\pm$ SD (standard deviation) from three independent biological replicates. Statistical significance (****, $P < 0.0001$) was assessed via an unpaired Student's *t* test. (C) The phenotypes of *E. coli* strains coproducing UreB with different combinations of accessory proteins were tested on Christensen's urea agar.

reconstituted strain resulted in a survival rate of 20% (Fig. 5B). No significant difference in virulence was found between the reconstituted and wild-type isolates. A histopathologic analysis of organs indicated that the fungus had almost been cleared from the lungs and livers of the surviving mice that were infected with Δ*ureB*. The histopathology of lungs and livers infected with the wild-type and complemented strains displayed evidence of invasive hyphal growth (Fig. 5C). Quantifying the area of fungal infiltration of the lungs and livers showed a significant difference between the wild-type, complementation, and Δ*ureB* mutant strains, concerning the fungal burden (Fig. 5D and E). Thus, we conclude that the absence of urease attenuates the virulence of *A. fumigatus* in both of the murine models of invasive infection that were used.

**Urease-mediated inhibition of acidification reduces the killing of conidia by macrophages.** To investigate the reason for the attenuated virulence, we first assessed the spore-killing efficiency of RAW 264.7 macrophages. The Δ*ureB* mutant showed significantly lower survival after being challenged with macrophages, in comparison to the wild-type. While the survival rate of the complemented strains were similar to that of the wild-type (Fig. 6A and B), we suspected the absent pH-modulating activity of urease to be responsible for the increased killing of the deletion mutant. Therefore, we measured the amount of conidia residing in acidified compartments using LysoTracker Red. The stained conidia of the wild-type, Δ*ureB* mutant, and complemented strains were incubated with RAW 264.7 macrophages. An analysis of colocalized conidia (blue) with acidified compartments (red) revealed that the wild-type is able to prevent acidification. In contrast, the deletion mutant showed a significantly higher acidification rate, whereas the complemented strain was comparable to the wild-type (Fig. 6C and D). In order to influence the pH of the intracellular compartments, the conidia should have active urease. We confirmed this by measuring the urease activity of protein extracts from resting conidia harvested from MAG medium. The wild-type and the complemented strain showed significantly higher urease activity of conidia, in comparison to the urease deletion strain (Fig. 6E). In addition, conidia from mycelium grown on AMM urea plates showed increased survival after coincubation with macrophages, in comparison to conidia from AMM ammonium plates (Fig. S17).

**FIG 3** Legend (Continued)
confirmed via Western blots that were probed with the anti-CBP antibody (bottom). The TAP fusion proteins are marked with asterisks. (B) Urease and the associated proteins were detected via a mass spectrometry analysis of whole lanes from silver-stained gels. Proteins found (√) or not found (○) in each fraction are shown in the table. (C) Urease activities of final purified fractions from TAP-tagged strains. The data that are shown are the mean values of assays performed in triplicate. Error bars denote standard deviations.

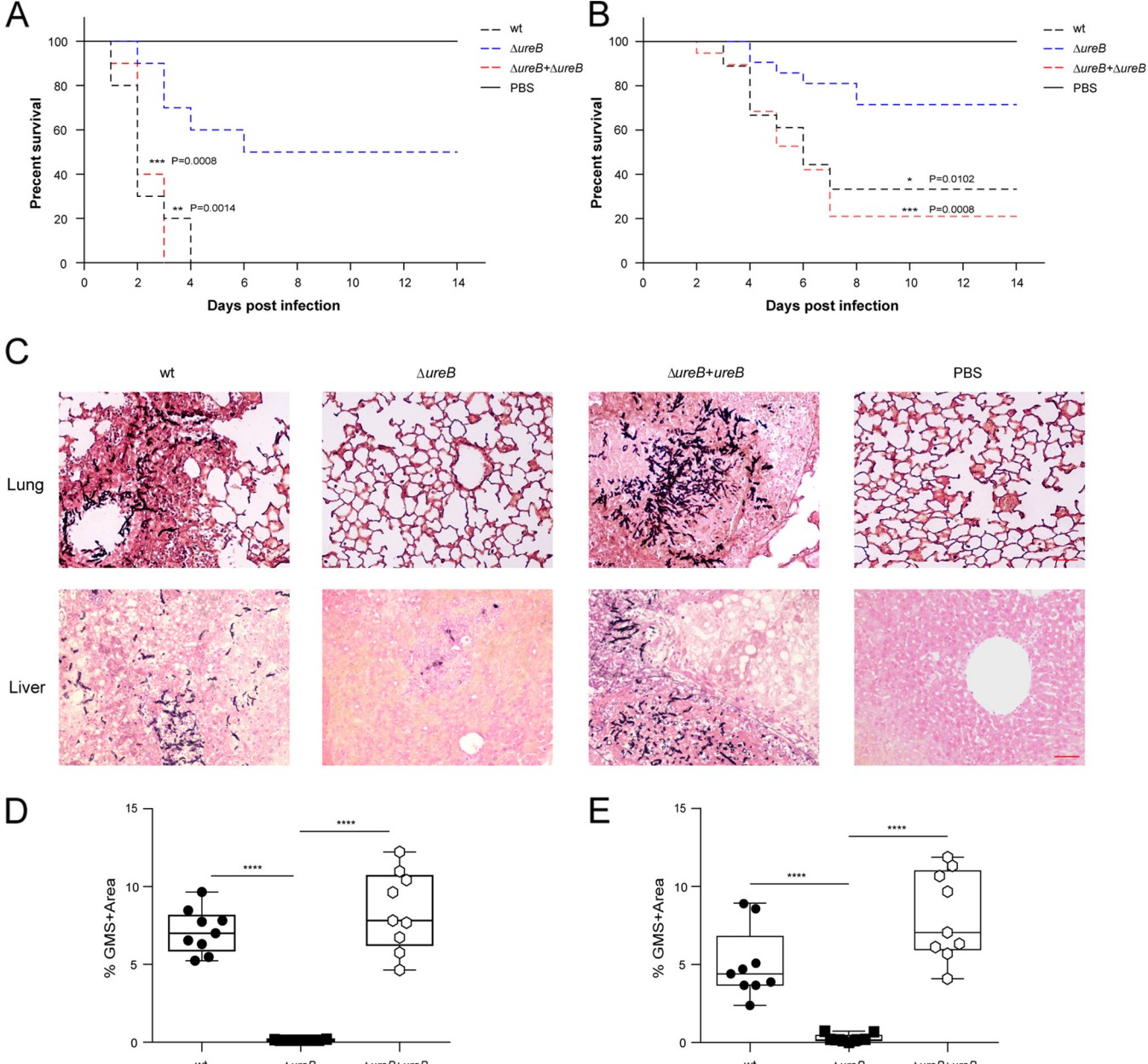

**FIG 5** The deletion of *ureB* attenuates the virulence of *A. fumigatus* in murine infection models. (A and B) Survival curves of cortisone acetate-treated mice that were infected with conidia from wild-type, Δ*ureB*, and Δ*ureB*+*ureB* strains via intratracheal instillation (A) or intravenous injection (B). The comparisons of the survival curves for Δ*ureB* versus the wild-type or Δ*ureB*+*ureB* were performed using the log-rank test (Mantel-Cox). (C) Histopathology of lungs from the intratracheal infection model and livers from the intravenous infection model. Representative lung and liver sections that were stained with GMS (Grocott's methenamine silver) exhibit fungal colonization (shown as black hyphae). Scale bar, 50 $\mu$m. (D and E) The fungal burdens of the lungs (panel D) and livers (panel E) were determined via the quantification of GMS staining in histological sections using ImageJ software. Statistical significance was assessed using Mann-Whitney tests. ****, $P < 0.0001$.

**The nickel chelator dimethylglyoxime (DMG) inhibits *A. fumigatus* growth on AMM medium.** After establishing the role of urease for the virulence of *A. fumigatus*, we wanted to evaluate whether it could be a potential drug target. Urease requires nickel for activity. Therefore, we tested the ability of the nickel chelator dimethylglyoxime (DMG) to inhibit the growth of *A. fumigatus* on AMM solid medium. A concentration of 5 mM DMG inhibits growth on AMM-nitrate medium and on AMM-urea. However, the colony diameter is smaller on 5 mM DMG AMM-urea, in comparison to 5 mM DMG AMM-nitrate. The growth inhibition could be reverted through the addition of nickel to the medium (Fig. 7).

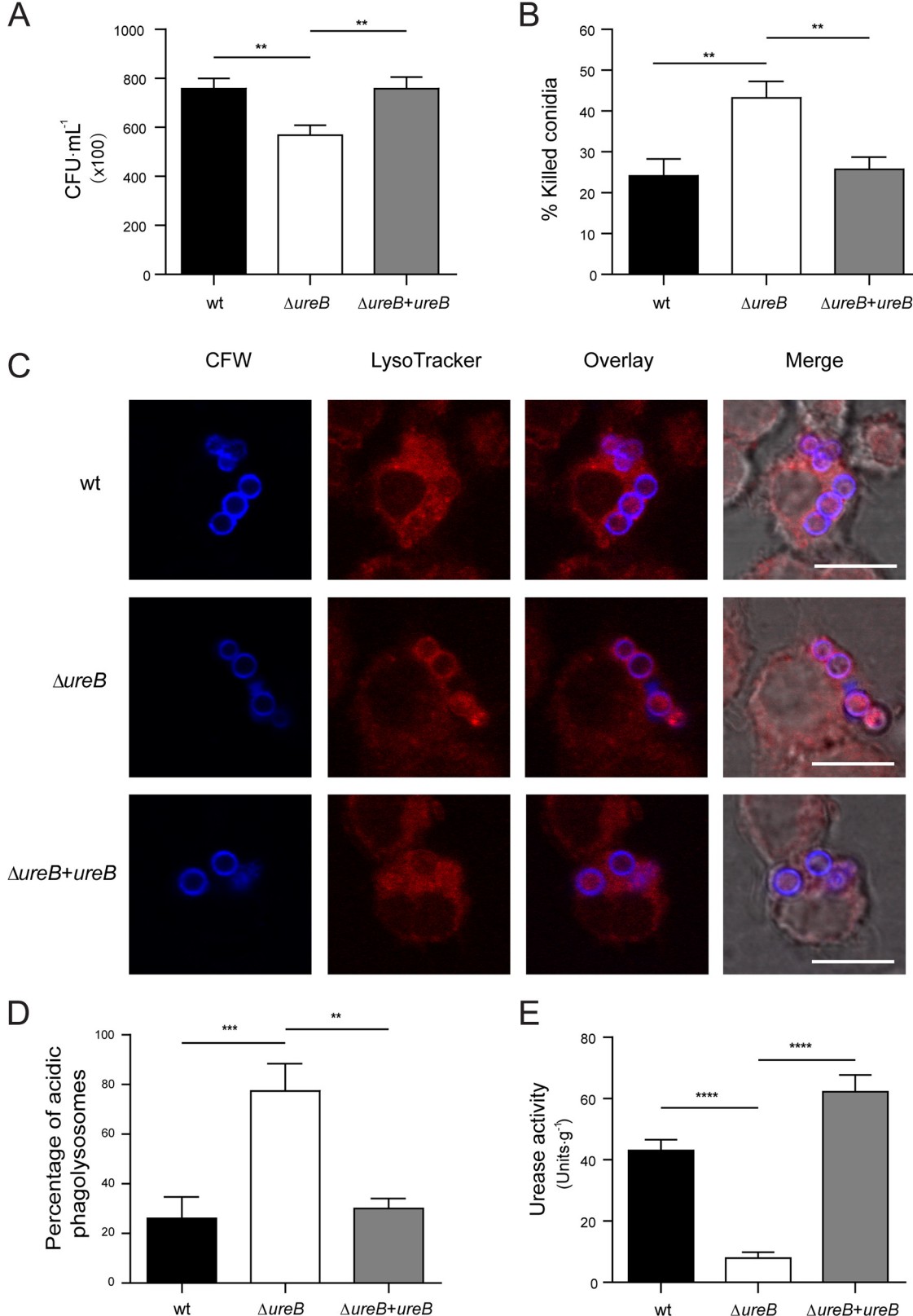

**FIG 6** The deletion of *ureB* stimulates enhanced fungal killing by RAW 264.7 macrophages and increases the amount of conidia in acidified compartments. (A and B) The fungal killing capacity of macrophages was assessed as the colony forming units (CFU) of remaining viable conidia (panel A) and the percentage of killed conidia (panel B) after 4 h of infection with a MOI of 2. (C) Colocalization of calcofluor white (CFW)-labeled conidia with acidified compartments in RAW 264.7 macrophages using LysoTracker

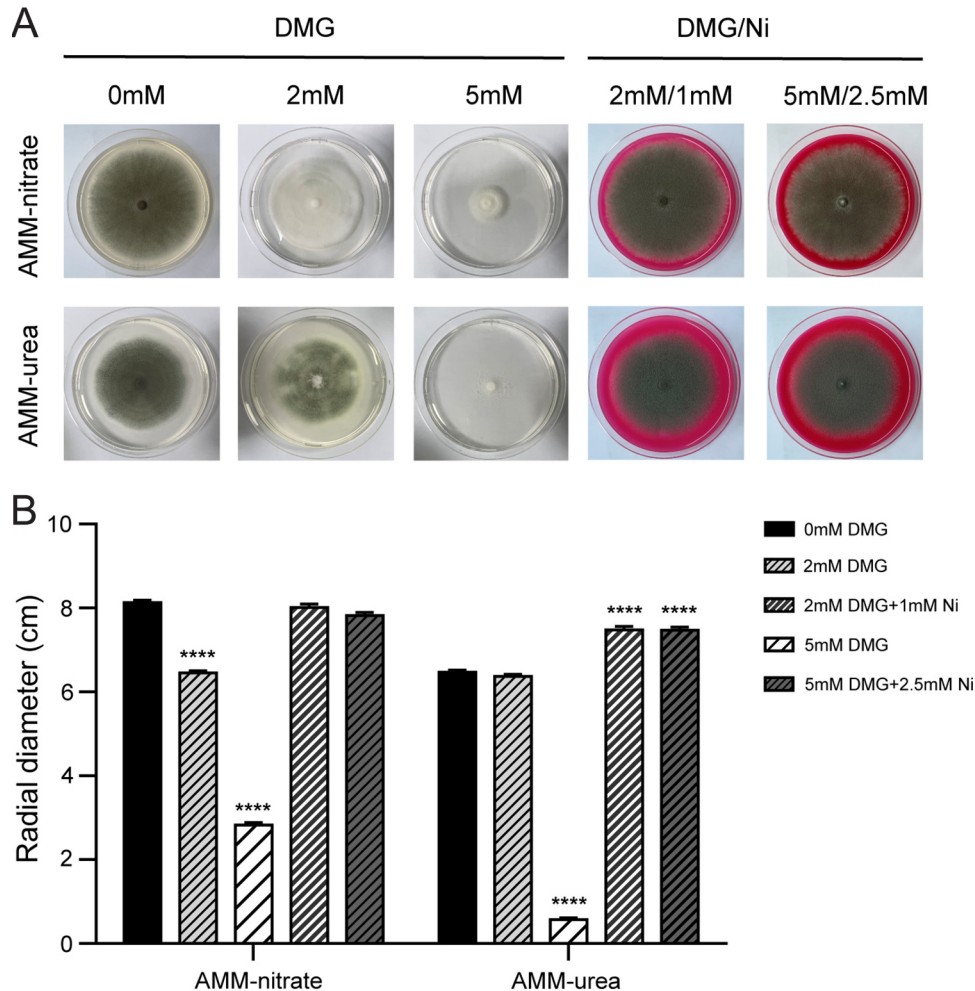

**FIG 7** Effect of the nickel chelator dimethylglyoxime (DMG) on the growth of *A. fumigatus*. (A) Growth phenotype of *A. fumigatus* on AMM-nitrate and AMM-urea agar in the presence of DMG or DMG plus stoichiometric amounts of nickel. (B) The radial diameter was measured after *A. fumigatus* was grown from $5 \times 10^3$ spores for 7 days at 37°C. The results represent the mean ± SD (standard deviation) from three independent biological replicates. Statistical significance (****, $P < 0.0001$) was assessed via an unpaired Student's *t* test.

**DMG enhances conidial killing by facilitating acidification and thereby increases survival in cortisone-acetate-treated murine infection models.** To explore the impact of DMG on the killing of conidia by macrophages, we first established which concentration was suitable without affecting the cell viability of the macrophages. We found that a concentration of 1 mM DMG does not affect macrophages (Fig. S19). Therefore, we used this concentration in subsequent experiments. The preincubation of the culture medium with 1 mM DMG led to elevated conidial killing of the *A. fumigatus* conidia, in contrast to the killing of *E. coli* and *C. albicans*, which are not affected by DMG (Fig. 8A). Furthermore, a significantly higher number of conidia were found to be residing in acidified organelles with DMG treatment, whereas the number of $SiO_2$ particles residing in acidified compartments is not impacted by DMG (Fig. 8B and C). To investigate whether the *in vitro* effect of DMG on acidification translates to *in vivo* significance, we repeated the lung and systemic

**FIG 6** Legend (Continued)
Red-DND99. The colocalization of conidia (blue) and acidified compartments (red) was analyzed microscopically after 2 h of coincubation with a MOI of 2. Scale bar 10 $\mu$m. (D) The percentages of conidia in acidic compartments. (E) Conidial urease activities were measured for wild-type, $\Delta ureB$, and complemented strains that were grown on MAG agar. The data represent the mean results and SDs (error bars) of three independent experiments. Statistical significance was assessed using one-way ANOVA tests with multiple comparisons. **, $P < 0.01$; ***, $P < 0.001$; ****, $P < 0.0001$.

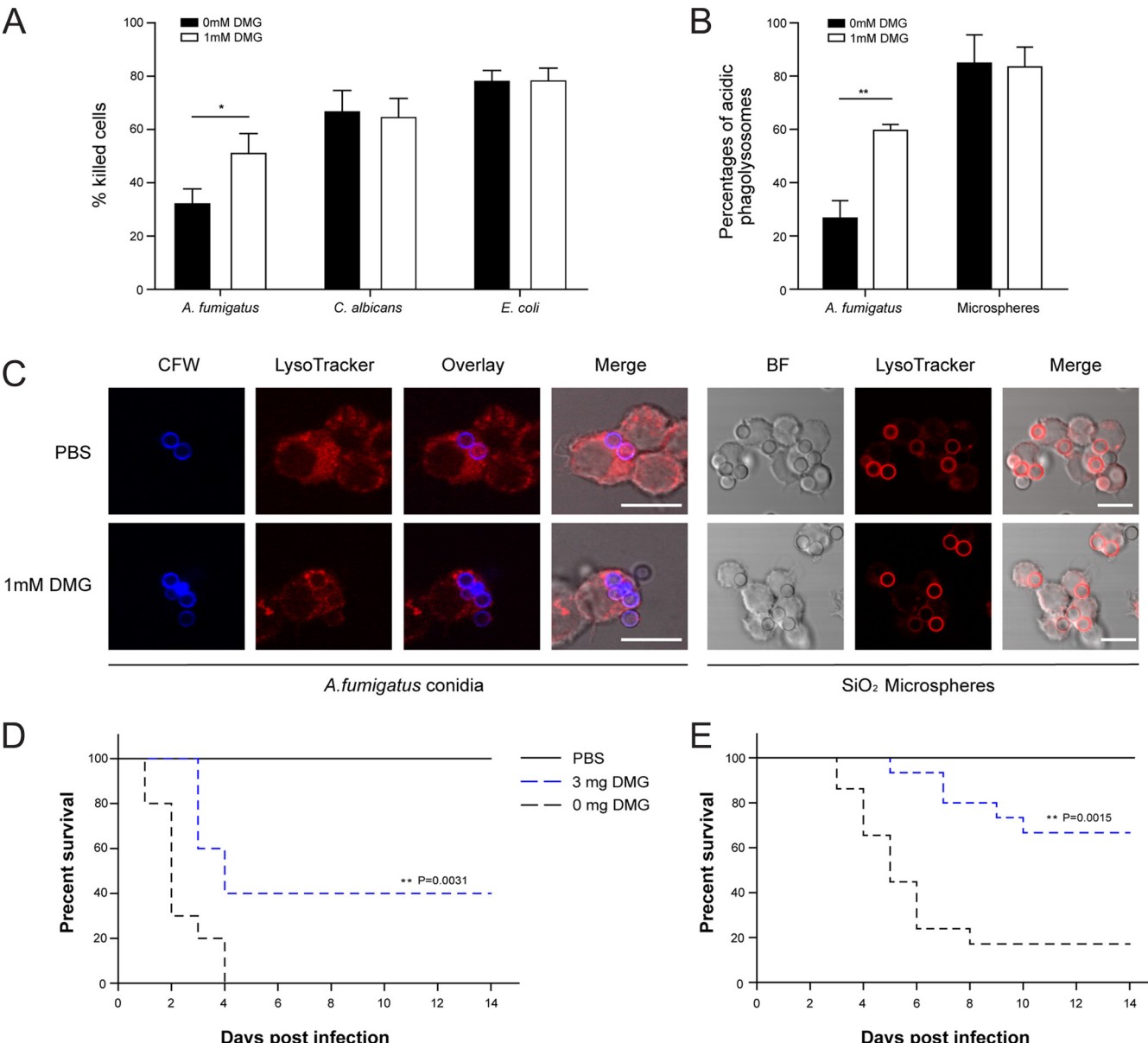

**FIG 8** Effect of the nickel chelator dimethylglyoxime (DMG) on *A. fumigatus* virulence. (A) Percentage of killed cells after coincubation with macrophages in DMEM pretreated with 1 mM DMG. (B) The percentages of conidia and SiO$_2$ beads in acidic compartments. The data the represent mean results and SDs (error bars) of three independent experiments. Statistical significance (*, $P < 0.05$; **, $P < 0.01$) was assessed via an unpaired Student's $t$ test. (C) Colocalization of calcofluor white (CFW)-labeled conidia or SiO$_2$ beads with acidified compartments in RAW 264.7 macrophages after DMG treatment. Scale bar, 10 $\mu$m. (D and E) Impact of DMG treatment on *A. fumigatus* virulence in murine intratracheal (panel D) and intravenous (panel E) infection models. Comparisons of survival curves were performed using the log-rank test (Mantel-Cox).

infection models with wild-type conidia but started to orally administer 3 mg of DMG daily. After 2 weeks, we observed significantly higher survival in the mice receiving DMG, in comparison to the group that was administered PBS (Fig. 8D and E). Also, the histopathology of the lungs and livers from the respective infection model showed that the organs from the mice that were treated with DMG showed complete clearance of fungal growth (Fig. S18).

## DISCUSSION

The *A. fumigatus* genome encodes three accessory proteins for the maturation of urease, which is consistent with other eukaryotes from yeast to plants. The conservation of these proteins is especially high in the regions related to nickel transfer, GTP binding, and protein-protein interactions. All eukaryotic urease maturation pathways lack a

homolog of the bacterial UreE, which serves as a bridge to acquire nickel from HypA, which is subsequently donated to UreG (19, 25). In *A. fumigatus*, the function of UreE seems to be adopted by UreG, based on the presence of an N-terminal extension that contains a His-rich hypervariable region that is followed by a highly conserved HXH motif (Fig. S6). This feature is only known from the C terminus of UreE from bacteria (Fig. S7) and not from the bacterial UreG orthologs (13, 28). Furthermore, UreG possesses highly conserved regions that are involved in GTP hydrolysis and nickel transfer to UreF/D (17, 52). It is reasonable to speculate that the extended N terminus of *A. fumigatus* UreG is essential for nickel transfer from a yet unknown source, as was demonstrated for the Ure7/UreG from *Cryptococcus* and *Arabidopsis* (28, 30).

Analogous to a previous report on *A. thaliana* urease (26), we showed that the accessory proteins UreD, UreF, and UreG of *A. fumigatus* are required to form active urease in the heterologous host *E. coli*. According to urease activation models that have been proposed for bacteria and plants, UreD is the first protein to bind to apourease (27, 28, 53). Our Strep pulldown results show that a direct interaction with UreB was observed only for UreD and not for UreF or UreG, further corroborating this finding in fungi. A UreD-UreF platform that is formed by the binding of UreF to UreD has been proven to be structurally essential for the following recruitment of UreG (17, 28). However, in our pulldown assay, UreG could bind to UreD without UreF, implying a novel interaction mode between the accessory proteins UreD, UreF, and UreG of *A. fumigatus*. Interestingly, a yeast two-hybrid (Y2H) analysis of the *C. neoformans* urease system indicated that the urease protein Ure1 interacts with all of the accessory proteins Ure4, Ure6, and Ure7 (homologs of bacterial UreD, UreF, and UreG); however, the interaction between Ure1 and Ure4 is weak and hard to detect (30). Our results combined with the Y2H data for *C. neoformans* suggest that the specific urease activation process in fungi might be different from the models proposed so far, albeit more evidence need to be provided.

Biochemical data from plants and bacteria demonstrated that the accessory protein UreG can bind to urease apoprotein in the presence of UreF/UreD to form a stable urease activation complex either *in vitro* or *in vivo* (17, 28). In this study, we were able to identify the interactions between UreB, UreD, and UreF in *A. fumigatus*. The fraction from the *ureG::cTap* strain showed only the ureG-cTap signal in both silver staining and MS identification and showed no urease activity. Of note, the fractions that were purified by the TAP-tagged version of UreB, UreD, and UreF showed different levels of urease activity, which implies that these fractions might contain urease that has been activated to a different extent. Considering the involvement of UreG in the activation of *A. fumigatus* urease in *E. coli*, we speculate that the *in vivo* interaction of UreG with the other proteins may be more transient and that the corresponding complexes are not stable enough for copurification. Thus, a complete model of nickel transfer during urease maturation in *A. fumigatus* and other fungi remains to be fully elucidated. Besides the urease maturation process, it is unknown whether there is direct contact of UreG with the nickel uptake system. Other factors could be involved, similar to the hydrogenase maturation factors HypA and HypB, which are responsible for the delivery of nickel to UreE in bacteria (24, 25, 54, 55). It is noteworthy that nickel is a transition metal, like iron. Highly sophisticated mechanisms for the intracellular homeostasis maintenance of iron to balance its essentiality, low environmental availability, and cytotoxicity have been elucidated in *A. fumigatus* cells (8, 56). It does not seem too farfetched that the acquisition, trafficking, and storage of nickel must be regulated in a similar way to allow for delivery to nickel-dependent proteins while avoiding toxic side effects.

Nitrogen catabolite repression (NCR) is a global regulatory mechanism that ensures the optimal utilization of nitrogen resources (57). In *A. nidulans*, NCR is mediated by the GATA-type transcription factor (TF) AreA interacting with the corepressor NmrA. When the preferred nitrogen sources, such as ammonium and glutamine, are lacking, NmrA dissociates from AreA, thereby allowing for the interaction of AreA with a substrate pathway-specific TF to activate the pathway that is required for the available

nonpreferred nitrogen sources (e.g., nitrate or urea) (58). A homolog of AreA also exists in *A. fumigatus* and functions as a global regulator of NCR. The disruption of *areA* in *A. fumigatus* causes a growth defect on a wide array of nitrogen sources, except for ammonium or glutamine (59). In the present study, urease and the accessory genes were determined to be essential for the growth of *A. fumigatus* with urea as the sole nitrogen source, and an increase of urease on both the mRNA and protein levels was observed after growth in a urea medium. Our results suggest a derepression of the genes that are required for urea degradation in *A. fumigatus* that is probably under the control of AreA.

Recently, *H. pylori* urease has been identified as a potential drug target for combating stomach infections. In response to the acidification of the periplasm of *H. pylori*, a transporter takes up urea into the periplasm, where urease hydrolyses the urea to produce ammonia and bicarbonate, which helps to neutralize the pH. This mechanism is crucial for *H. pylori* to survive and cause infection in the stomach (60, 61). The role of the *A. fumigatus* urease is somewhat similar to that of the *H. pylori* enzyme. We have demonstrated that a *ureB* deletion mutant is reduced in its ability to counteract phagosome acidification, which leads to higher killing rates and a reduced virulence in murine infection models. This is consistent with the proposed role of UreB in the production of ammonia, which could neutralize the initial acidification of the phagosome, which is a crucial step for fusion with the lysosome and downstream killing (62). A similar effect on the inhibition of acidification was described for the *pksP* mutant, which was later shown to be linked to the dysregulation of membrane microdomains by melanised conidia, which prevent the assembly of ATPase and subsequent acidification (63–65). Although urease has been identified as virulence factor in several pathogens, here, we show for the first time the importance of urease for a filamentous human fungal pathogen.

The search for new antifungal drugs is based on the principles that the target should be present in a wide range of pathogenic fungi and absent or less susceptible in the host. Previous studies have already shown that the nickel-dependent urease is present in basidiomycetes and ascomycetes besides hemiascomycetes (66). We concluded from our bioinformatics analyses that antifungals directed against UreB have the potential to be selective; however, they may be limited in their spectra of activity. The usefulness of narrow spectrum antifungals should not be underestimated. Advanced methods for the detection of fungal pathogens allow for identification on the species level, and different drugs are already preferentially administered, based on the pathogen and the site of infection. For example, the echinocandins are increasingly used for the treatment of disseminated candidiasis; however, the azoles are preferred for the treatment of aspergillosis (67). Especially, nickel chelation therapy could be interesting, as this metal seems to be inessential for human cells but essential for the activity of certain enzymes from pathogens. We were able to observe an inhibitory effect of DMG on AMM nitrate medium. A possible explanation would be that *A. fumigatus* contains other nickel-dependent proteins that are independent of urea metabolism. Possible candidates for such proteins could be nickel-dependent superoxide dismutase, glyoxalase, or acireductone dioxygenase, and these could potentially make nickel-chelation therapy more effective (68). Furthermore, the already established efficiency of FDA-approved compounds as urease inhibitors and the constant development of novel inhibitors could allow for the fast adaptation of these molecules to inhibit fungal pathogens (69).

In conclusion, the formation of active urease, which is mainly localized in the cytoplasm, in *A. fumigatus* requires the assistance of three maturation proteins (Fig. 9). The urease transcript and protein levels are induced after growth on media with urea as the sole nitrogen source but are not completely repressed when nitrate is the nitrogen source. The activity of urease can be detected in resting conidia and counteracts the acidification of the phagosome after phagocytosis in macrophages. This allows for the survival of conidia and decreases the virulence of a urease deletion mutant in murine infection models. We were able to show that the nickel chelator dimethylglyoxime can inhibit the growth of *A. fumigatus*. DMG enhances the killing by macrophages and

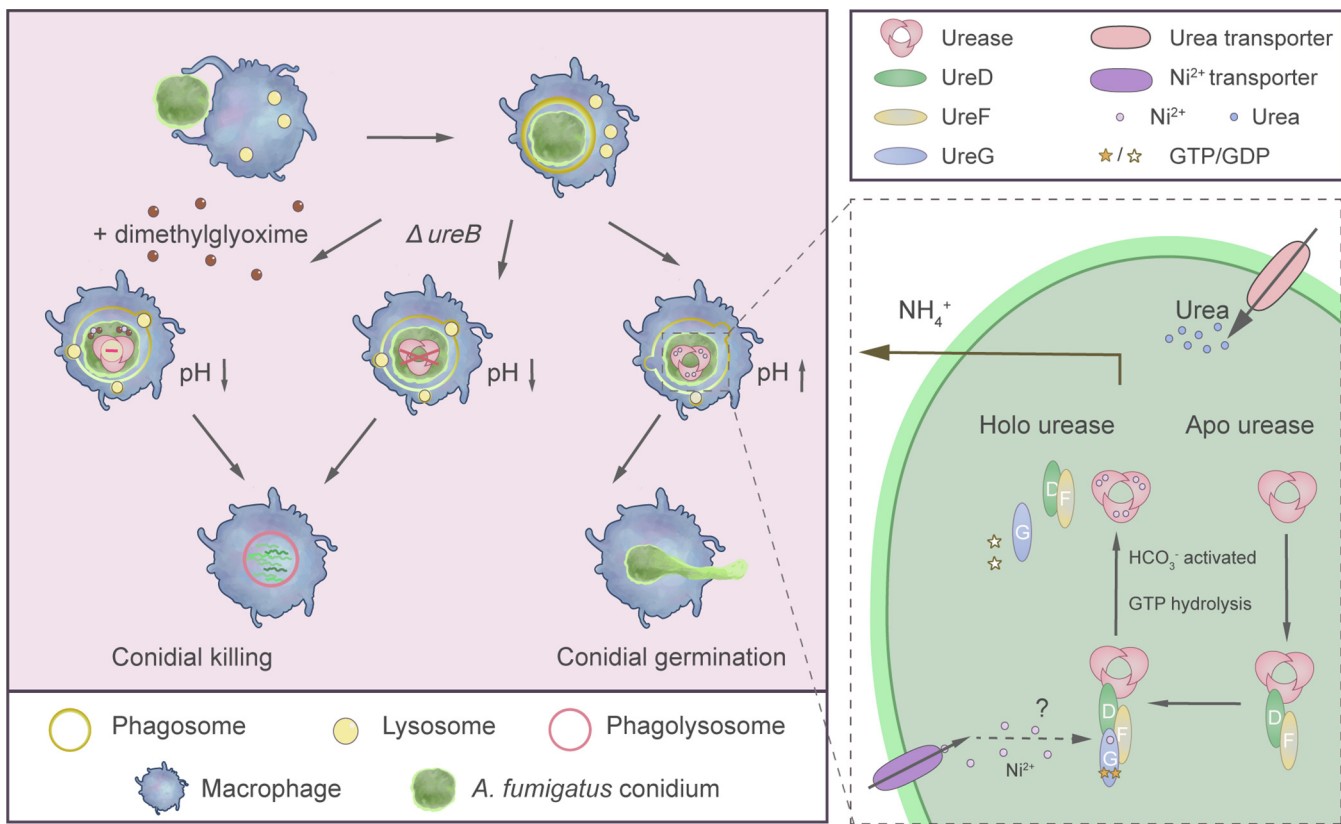

**FIG 9** The role of urease in *A. fumigatus* virulence (left panel) and the urease assembly model for *A. fumigatus* (right panel). After the phagocytosis of conidia by macrophages, the activity of urease helps to prevent the acidification of the phagosome. Therefore, the conidia cannot be efficiently killed and start to germinate. The deletion of the urease gene or treatment with DMG allows for the acidification and subsequent killing of the conidia. The assembly of active urease in *A. fumigatus* is dependent on three accessory proteins, which help to install the nickel in the active site of urease (49).

increases the survival rate in murine infection models. Therefore, nickel-chelators or other urease inhibitors are interesting targets for antifungal therapy and should be considered for further investigation. Many of the antifungals that are used to treat invasive aspergillosis have significant pharmacological shortcomings, and we see an increase in resistant clinical isolates. This and the high mortality rates that are associated with systemic fungal infections call for new druggable targets to fight fungal infections. In this study, we have demonstrated the importance of urease for *A. fumigatus* virulence and showed that nickel chelators or substrate analogues have the potential to inhibit fungal growth.

## MATERIALS AND METHODS

**Ethics statement.** All animal procedures were performed in accordance with the Guidelines for Care and Use of Laboratory Animals of Zhejiang University and approved by the Animal Ethics Committee of Zhejiang University (protocol number ZJU20220150).

**Fungal strains and cultivation conditions.** The fungal strains that were used in this study are listed in Table 1. The plasmids that were used in this study are listed in Table S1. The primers that were used in this study are listed in Table S2. *A. fumigatus* CEA17Δ*akuB*$^{KU80}$ (49, 50) was used as the wild-type parent for all of the fungal strains that were constructed in this study. *A. fumigatus* strains were grown at 37°C with continuous shaking (200 rpm) in *Aspergillus* minimal medium (AMM) (70) or at 37°C on AMM agar (AMM with 1.5% wt/vol agar). Urea (5 mM) was added to AMM to replace nitrate as the sole nitrogen source, as needed. For the formation of conidia, *A. fumigatus* was cultivated on MAG (71) agar plates at 37°C for 4 days. Conidia were harvested in 0.9% (wt/vol) NaCl/0.1% (vol/vol) Tween 80 and were counted using a cell counting chamber. AMM agar plates with either hygromycin (200 $\mu$g · mL$^{-1}$) or pyrithiamine (0.1 $\mu$g · mL$^{-1}$) were used to select fungal transformants. Christensen's urea agar medium (peptone 1 g · L$^{-1}$, Glucose 1 g · L$^{-1}$, NaCl 5 g · L$^{-1}$, Na$_2$HPO$_4$ 1.2 g · L$^{-1}$, KH$_2$PO$_4$ 0.8 g · L$^{-1}$, phenol red 0.012 g · L$^{-1}$, 2% [wt/vol] urea, 1.5% [wt/vol] agar) was used for the phenotypic assays of urease activity. The plasmids pUChph and pSK275 were the sources of the hygromycin-resistant cassette (Hyg$^R$) and the pyrithiamine-resistant cassette (PtrA$^R$), respectively. The cTap and nTap were provided by plasmids pME2967 and pME2968. These plasmids

**TABLE 1** Fungal strains used during this study

| Strain[a] | Description[b] | Source |
|---|---|---|
| CEA17Δ*akuB*$^{KU80}$ | *akuB*$^{KU80}$::*pyrG*; PyrG$^+$ | 49 |
| Δ*ureB* | Δ*akuB*$^{KU80}$; *ureB*::*ptrA*; PtrA$^R$ | This study |
| Δ*ureB*+*ureB* | Δ*akuB*$^{KU80}$; Δ*ureB*::*ureB*; PtrA$^R$, Hyg$^R$ | This study |
| Δ*ureD* | Δ*akuB*$^{KU80}$; *ureD*::*ptrA*; PtrA$^R$ | This study |
| Δ*ureD*+*ureD* | Δ*akuB*$^{KU80}$; Δ*ureD*::*ureD*; PtrA$^R$, Hyg$^R$ | This study |
| Δ*ureF* | Δ*akuB*$^{KU80}$; *ureF*::*ptrA*; PtrA$^R$ | This study |
| Δ*ureF*+*ureF* | Δ*akuB*$^{KU80}$; Δ*ureF*::*ureF*; PtrA$^R$, Hyg$^R$ | This study |
| Δ*ureG* | Δ*akuB*$^{KU80}$; *ureG*::*ptrA*; PtrA$^R$ | This study |
| Δ*ureG*+*ureG* | Δ*akuB*$^{KU80}$; Δ*ureG*::*ureG*; PtrA$^R$, Hyg$^R$ | This study |
| EGFP::*ureB* | Δ*akuB*$^{KU80}$; EGFP::*ureB*::*hph*; Hyg$^R$ | This study |
| nTap::*ureB* | Δ*akuB*$^{KU80}$; nTap::*ureB*::*hph*; Hyg$^R$ | This study |
| *ureD*::cTap | Δ*akuB*$^{KU80}$; *ureD*::cTap::*ptrA*; PtrA$^R$ | This study |
| *ureG*::cTap | Δ*akuB*$^{KU80}$; *ureG*::cTap::*ptrA*; PtrA$^R$ | This study |
| nTap::*ureF* | Δ*akuB*$^{KU80}$; nTap::*ureB*::*ptrA*; PtrA$^R$ Hyg$^R$ | This study |

[a]All strains are *A. fumigatus* strains, unless indicated otherwise.
[b]PtrA, pyrithiamine; Hyg, hygromycin.

were preserved in *E. coli* cells that were grown at 37°C in either Luria-Bertani broth or agar supplemented with 100 $\mu$g · mL$^{-1}$ ampicillin.

**Identification of target genes.** The putative urease encoding gene *ureB* (AFUA_1G04560) and the accessory genes *ureD* (AFUA_2G16070) and *ureG* (AFUA_2G12900) were identified from the FungiDB database, which is available at https://fungidb.org/fungidb/app/. An ortholog of *ureF* was identified via a BLAST analysis against the *A. fumigatus* genome, using *A. thaliana* UreF as a query.

**Phylogenetic analysis.** Phylogenetic analyses were conducted using MEGA 11 (72). All sequences were aligned using the ClustalW software that is provided by MEGA 11. Phylogenetic trees were prepared via the maximum likelihood method, using the Jones-Taylor-Thornton (JTT) distance estimation models (73). Bootstrap values were calculated from 1,000 replications of the bootstrap procedure by using the programs within the MEGA 11 software package.

**Genetic manipulation on *A. fumigatus*.** Gene deletion constructs were generated using a modified PCR fusion approach (74). The deletion constructs for all target genes were created using the following steps. The 5′-flanking region was amplified with the primer pairs P_5′-1/P_5′-2, and the 3′-flanking region was amplified with theprimer pairs P_3′-1/P_3′-2. The ptrA resistance cassette was amplified from the plasmid pSK275 with the primers P_R-1/P_R-2. To facilitate gene fusion, the primers P_5′-2 and P_3′-1 included regions that were complementary to the terminal regions of the ptrA primers. All of the PCR fragments were purified via gel extraction. The final deletion construct was generated via a three-fragment PCR using the primers P_5′N/P_3′N. All of the PCRs were performed using Q5 High-Fidelity DNA Polymerase (NEB). The resulting PCR product was purified and transformed into *A. fumigatus* wild-type protoplasts as previously described (75). The complementation of the mutants Δ*ureB*, Δ*ureD*, and Δ*ureG* was done by following the method by D. Wartenberg, et al. (76). The 5′ UTR flanking region, in addition to the coding sequence, and the 3′ UTR flanking regions of *ureB*, *ureD*, and *ureG* were amplified from genomic DNA via PCR. The amplified fragments were then ligated into the plasmid pUChph at the XbaI or HindIII site upstream of the Hyg resistance cassette. The plasmids containing the cloned gene cassettes were transformed into the respective gene deletion strains to create the complemented strains. A fusion DNA fragment 5′arm::*ureF*::Hyg$^R$::3′arm was introduced to the Δ*ureF* deletion mutant to generate the reconstituted strain via homologous recombination.

For the construction of the nTap-tagged strains, the *ureB* and *ureF* genes were amplified from genomic DNA, and each was homologously integrated into the 3′ end of nTap in pME2968 using a One Step Cloning Kit (Vazyme), thereby generating pME2968_nTap-*ureB* and pME2968_nTap-*ureF*. The 5′arm and 3′ arm of *ureB* and the Hyg$^R$ cassette were amplified. The fragments, together with the plasmid-derived nTap::*ureB*, were fused to form the 5′arm::nTap::*ureB*::Hyg$^R$::3′arm long fragment, whereas the PtrA$^R$ cassette was used to make the 5′ arm::nTap::*ureF*::PtrA$^R$::3′arm fusion fragment. To construct the cTap-tagged strains, the PtrA$^R$ cassette was released via SpeI/PstI digestion and was inserted into pME2967 at the C terminus of cTap, thereby resulting in pME2967_cTap-PtrA$^R$. Then, *ureD* and *ureG*, including the 5′arms and the 3′arms of *ureD* and *ureG*, were amplified. These fragments were fused to the cTap::PtrA$^R$ module via PCR, yielding the 5′arm::*ureD*::cTap::PtrA$^R$::3′arm and the 5′arm::*ureG*::cTap::PtrA$^R$::3′arm fragments, respectively. Prior to the generation of the EGFP::*ureB* strain, the 5′arm of *ureB* was integrated into pFastBac_EGFP at the 5′ end of EGFP via homologous recombination, thereby generating the pFastBac_*ureB*5′arm-EGFP plasmid. Then, the *ureB*5′arm::EGFP, *ureB* coding region, the Hyg$^R$ cassette, and the 3′ arm of *ureB* were amplified and fused into the 5′arm::EGFP::*ureB*::Hyg$^R$:: 3′arm long fragment. All of the gene constructs were transformed into *A. fumigatus* protoplasts as previously described (75).

All of the mutant, complemented, and TAP/EGFP-tagged transformants were confirmed via PCR and were tested for growth and urease activity on AMM-urea and on Christensen's urea (using phenol red as a pH indicator) plates, respectively.

**Southern blot analysis.** The *A. fumigatus* mutant and reconstituted strains were verified via Southern blotting. Briefly, the genomic DNA of *A. fumigatus* was isolated using a DNeasy Plant Minikit (Qiagen), and

20 $\mu$g DNA were digested by specific restriction enzymes (TaKaRa, Japan). DNA fragments were separated on a 0.8% (wt/vol) agarose gel and were transferred onto a $N^+$ nylon membrane (Roche, Germany). The labeling of the DNA probe was performed using PCR DIG Labeling Mix (Roche, Germany). For the hybridization and detection of DNA-DNA hybrids, the DIG High Prime DNA Labeling and Detection Starter Kit II (Roche, Germany) was used, according to the manufacturer's instructions.

**RNA extraction and real-time PCR.** *A. fumigatus* mycelium was grown from $1 \times 10^6$ conidia in AMM at 37°C, 200 rpm for 16 h, and this was followed by a transfer to fresh AMM-urea medium for further cultivation. Mycelia were harvested, snap-frozen in liquid nitrogen, and ground via cryo-compaction using a ball mill (MeiBi). Total RNA was extracted using an RNeasy Plant Minikit (Qiagen), according to the manufacturer's protocol. Reverse transcription was performed using 10 ng total RNA, using the Prime Script RT Reagent Kit with gDNA Eraser (TaKaRa, Japan). Real-time PCR was conducted using TB Green Premix *Ex Taq* II (TaKaRa, Japan) on a Bio-Rad CFX96 real-time PCR system (Bio-Rad Laboratories), following the program: 95°C for 30 sec/40 cycles at 95°C for 5 sec/60°C for 30 sec. Samples were run in triplicate. The relative expression of transcripts was quantified via the comparative critical threshold (CT) value method and was normalized to the $\beta$-actin expression. For the PCR primers used, see Table S2.

**Protein extraction from *A. fumigatus*.** Mycelia were collected and ground into a fine powder in liquid nitrogen. The ground mycelia were thawed on ice and were resuspended in phosphate-buffered saline (PBS) (pH 7.4) containing 1 mM DTT and 2 mM PMSF. Crude extracts were centrifuged at $12,000 \times g$ for 5 min at 4°C to remove the insoluble fraction, and the protein concentration in the supernatant was measured using Bradford reagent (Sangon Biotech).

**TAP purification.** AMM liquid cultures were inoculated with $1 \times 10^6$ spores mL$^{-1}$ of *A. fumigatus* strains and were incubated at 37°C and 200 rpm for 16 h. The mycelia were shifted to new AMM-urea media and were grown for another 16 h. Tandem affinity purification was conducted as previously described (77), with slight modifications. Two grams of lyophilized mycelium were ground into a fine powder. The ground mycelia were extracted in buffer B250m and crude lysates were obtained via centrifugation at $40,000 \times g$ for 30 min at 4°C. The cleared supernatant was transferred onto a 10 mL chromatography column (Bio-Rad) that contained 300 $\mu$L of IgG Sepharose 6 Fast Flow (GE Healthcare), and it was incubated at 4°C on a rotator for 3 h. The IgG Sepharose was washed twice with 10 mL of buffer W250, once with 10 mL of buffer W150, and once with 10 mL of buffer TCB. TEV cleavage was performed under rotation using 20 $\mu$g of TEV (produced by our lab) overnight at 4°C. The eluate was incubated with 300 $\mu$L of CBB equilibrated calmodulin Sepharose 4B (GE Healthcare) for 1 h at 4°C on a rotator. The calmodulin Sepharose was subsequently washed three times with 1 mL of CBB with 0.02% Tergitol-NP-40, and it was finally eluted twice with 500 $\mu$L of CEB. This eluate was TCA precipitated, electrophoresed in a 10% Bis-Tris polyacrylamide gel, and stained with silver nitrate (78). The gel of the separation lane was cut out and submitted for the mass spectrometry identification of proteins.

**Enzyme activity assay.** 100 $\mu$g of protein were added to 200 $\mu$L of PBS (pH 7.4) with 50 mM urea and 5 $\mu$M NiCl$_2$, mixed well, and incubated at 37°C for 1 h. Subsequently, an Ammonia Assay Kit (Solarbio) was used to measure the production of ammonia. A 50 $\mu$L volume was removed from the reaction solution and mixed with 125 $\mu$L of extraction buffer. Supernatant was obtained via centrifugation at $12,000 \times g$ for 10 min. A volume of 100 $\mu$L was removed and placed into the wells of a 96-well plate, and Reagent 1 and Reagent 2 were added sequentially. The assay was incubated for 20 min at 37°C, and the $A_{630nm}$ was measured in a plate reader. The ammonia concentration was determined via the utilization of a standard curve (0.0625 mM to 8 mM NH$_4$Cl) that was established simultaneously with each assay and used to quantify the urease activity. One unit of urease catalyzes the liberation of 1 $\mu$mol of ammonia per min.

**Western blot analysis.** In order to analyze the expression level of EGFP-UreB and the enrichment of the TAP fusion proteins, the rabbit anti-GFP polyclonal antibody (1:1,000, GenScript) and the anti-calmodulin binding protein epitope tag antibody (1:5,000, UPSTATE) were used. The total proteins of *A. fumigatus* or the purified TAP fusion proteins were mixed with NuPAGE LDS sample buffer, separated on a 4 to 12% (wt/vol) Bis-Tris polyacrylamide gel (Invitrogen), and wet transferred to a PVDF membrane (Millipore). The membranes were blocked with 5% skim milk in TBST (0.05% Tween 20 in 25 mM Tris-HCl and 500 mM NaCl [pH 7.5]) for 2 h at room temperature and were then incubated for 1 h with the specific primary antibodies. The membranes were washed three times for 15 min with PBST and were further incubated with a 1:10,000 dilution of horseradish peroxidase (HRP)-conjugated goat anti-rabbit IgG (EarthOx) for 1 h. After 3 washes with TBST, the protein bands were detected using ECL chemiluminescence reagents (Coolaber).

**Construction of *E. coli* protein production plasmids.** The cDNA of UreB was amplified after the reverse transcription of *A. fumigatus* total RNA, which introduced an NdeI site at the start codon and an XbaI site at the stop codon. The *ureB* fragment was then cloned into the NdeI and XbaI sites of pnEA_vStrep. An in-frame translational fusion of the HA-tag at the C terminus of UreG was created via PCR, and the resulting *ureG::ha* sequence was subsequently integrated into ppCS at the multiple cloning sites (between NdeI and XbaI) via homologous recombination. These steps generated pnEA_vStrep-UreB and ppCS_UreG-HA. After amplification, the *ureG::ha* that was flanked by IPTG-induced T7 transcription elements was homologously integrated into pnEA_vStrep-UreB downstream of the stop codon of *ureB*, thereby resulting in the coexpression plasmid pnEA_vStrep-UreB/UreG-HA (1). The UreG-free control plasmid pnEA_vStrep-UreB-HA (2) was also constructed through a PCR-mediated HA-tag introduction and homologous integration, based on overlapping primer pairs. By inserting the UreD cDNA into pMalC2HTEV, which contained the open reading frame of MBP, via the NcoI and HindIII sites, the *mbp::ureD* fusion was obtained and was then cloned into the NdeI and NheI sites of ppCS, thereby generating ppCS_MBP-UreD (3). The UreF coding cDNA was amplified via PCR, which introduced a His tag at the 5' end of the start codon, and cloned into ppCS via the BamHI and XbaI sites. This resulted in

ppCS_His-UreF. Subsequently, the MBP-UreD expression cassette was released from ppCS_MBP-UreD by SpeI and NheI, and it was ligated into the SpeI-digested and dephosphorylated ppCS_His-UreF to create the ppCS_MBP-UreD/His-UreF (4) coexpression plasmid. Using this plasmid as a template, ppCS_MBP/His-UreF (5) (lacking the UreD coding sequence) was produced by inserting a stop codon at the 3′ end of the *mbp* gene and following this with self-ligation, based on the bilateral *kpn*I sites that were introduced via PCR. For the existence of a His tag between MBP and UreD, ppCS_MBP-UreD (3) could be directly used as a UreF-null plasmid. All of the constructs were confirmed via sequencing when PCR steps were involved. The plasmids pnEA_vStrep and ppCS possess the compatible origins of replication ColE1 and CDF, thereby allowing for the simultaneous presence of their derivatives in the same cell.

**Coexpression and Strep pulldown assay.** The plasmid combinations (1) + (4), (1) + (5), (1) + (3), and (2) + (4) that were constructed above were, respectively, cotransformed into *E. coli* BL21(DE3) pLysS, thereby allowing for the coexpression of UreB with all three or with any two of the accessory proteins. All *E. coli* transformants were cultured in Luria Broth (LB) medium (100 $\mu$g · mL$^{-1}$ ampicillin, 50 $\mu$g · mL$^{-1}$ streptomycin, and 50 $\mu$g · mL$^{-1}$ chloramphenicol) at 37°C overnight. The bacterial cultures were then diluted 1:100 into 1 L of LB medium that contained 100 $\mu$M NiCl$_2$ and proper antibiotics for 2 h of subculture at 37°C. The expression of UreB and accessory proteins was induced by adding 1 mM isopropyl $\beta$-D-1-thiogalactopyranoside (IPTG) when the OD$_{600}$ reached 0.6 to 0.8. After 5 h of induction at 30°C, the cells were harvested by centrifugation and resuspended in Buffer W (100 mM Tris-HCl, pH 8.0; 150 mM NaCl; 1 mM PMSF as a protease inhibitor), and this was followed by lysis on ice via sonication. The lysates were centrifuged (12,000 × *g*, 20 min) at 4°C to remove insoluble cell debris, and they were subjected to further filtration using a Millex-HA filter (0.45 $\mu$m, Millipore). The filtrates were loaded onto Strep-Tactin XT columns (2 mL bed volume) that had been preequilibrated with Buffer W, and this was followed by washing with 5 column volumes of the same buffer. Strep-tagged UreB and its associated accessory protein(s) were eluted with Buffer W supplemented with 50 mM biotin and concentrated by using a 10,000-molecular-weight-cutoff Amicon Ultra centrifugal filter device (Millipore). The final eluates were separated by 12% SDS-PAGE, and the presence of UreB and the accessory proteins was substantiated by Western blotting as stated above, using mouse antibodies against Strep (1:5,000, Immunoway), MBP (1:1,000, GenScript), His (1:5,000, EarthOx) and HA (1:5,000, GenScript), respectively. An anti-mouse IgG-HRP secondary antibody (1:10,000, EarthOx) was used for visualization.

**Macrophage killing assay.** To look into the intracellular persistence of *A. fumigatus* conidia, *E. coli*, and *C. albicans* after phagocytosis by macrophages, an intracellular killing assay was conducted. *A. fumigatus* strains were streaked onto MAG media or AMM agar with different nitrogen sources (nitrate, urea, or ammonium tartrate). The conidia were harvested after 4 days of growth at 37°C. The *E. coli* BL21(DE3) and *C. albicans* (Bio-52835) were harvested after 18/48 h of growth on LB at 37°C and YPD media at 30°C. RAW 264.7 macrophages were bought from the National Collection of Authenticated Cell Cultures (SCSP-5036) and were cultivated in Dulbecco's modified Eagle's medium (DMEM; Gibco) supplemented with 10% (vol/vol) fetal bovine serum (Gibco), and 1% (wt/vol) GlutaMAX (Gibco) at 37°C and 5% CO$_2$. Cell numbers were determined using a Neubauer chamber. 200 $\mu$L of DMEM-10% FBS that contained 5 × 10$^4$ RAW 264.7 macrophages and 1 × 10$^5$ *A. fumigatus* conidia/*E.coli*/*C. albicans* (multiplicity of infection [MOI] 1:2) were added in 96-well microplates. After 4 h of coculture at 37°C with 5% CO$_2$, the plates were centrifuged (400 × *g* for 10 min), the supernatants were discarded, and the pellet was retained in individual tubes. Phagocytized conidia were released by lysing the macrophages with sterilized ice-cold water. Cell lysates were collected in the same individual tubes that contained the pellets. The combined samples, which contained both ingested and noningested cells, were serially diluted and plated onto MAG/LB/YPD plates. CFU were counted after 18 h of incubation at 37°C or 30°C. All assays were performed with three wells in three independent experiments.

**Phagolysosomal acidification assay.** The acidification of the phagosome was assessed via the use of the acidotropic dye LysoTracker Red DND-99 (Thermo Fisher Scientific). RAW 264.7 macrophages, cultured overnight on coverslips in a 24-well plate, were prestained with 50 nM LysoTracker for 1 h at 37°C in a CO$_2$ incubator. Fresh *A. fumigatus* conidia that had been stained with 100 $\mu$g · mL$^{-1}$ calcofluor white and Uniform Silica Microspheres (Aladdin) were used to challenge the prestained cells at an MOI of 2. Meanwhile, another aliquot of 50 nM LysoTracker was added to the cells. The plates were centrifuged for 5 min at 100 × *g* and 37°C to synchronize phagocytosis. After 2 h of coincubation at 37°C with 5% CO$_2$, the cells containing conidia were washed once with PBS and were fixed for 10 min at room temperature with 4% (vol/vol) formaldehyde. The cells were washed 3 times for 5 min with PBS, and the coverslips were mounted on glass slides and examined under the confocal laser scanning microscope (CFLSM; Olympus, Japan). The images (150 for each group) were analyzed using FV1000 software (Olympus, Japan). Conidia that were surrounded by a ring-like or round-like LysoTracker signal were considered to be trapped in an acidified phagolysosome and were counted as acidified conidia. The ratio of acidified conidia to the total ingested conidia number was calculated to give the acidification ratio. All values represent the mean results ± standard deviations (SD) of three experiments.

For the DMG experiments, before the infection of the cells by *A. fumigatus* conidia, DMG was mixed with DMED to a final concentration of 1 mM and was incubated for 1 h at 37°C in a CO$_2$ incubator.

**Cell viability.** 5 × 10$^3$/100 $\mu$L RAW 264.7 macrophage cells were inoculated per well in a 96-well plate, and they were incubated in an incubator (37°C and 5% CO$_2$) overnight. The medium was exchanged with preincubated DMEM with different concentrations of DMG (0 to 10 mM), and it was incubated in an incubator for 4 h. Then, 10 $\mu$L Cell Counting Kit-8 (Coolaber, SK2060) were added into each well, slightly mixed, and incubated in a cell incubator for 1 h. The A$_{450nm}$ value was measured using a microplate reader.

**Murine infection models.** *A. fumigatus* strains were grown on malt extract agar (MAG) at 37°C for 4 days. Fresh conidia were collected via flooding with PBS-T and counted using a Neubauer chamber. The

viability of the administered inoculum was determined by incubating a serial dilution of the conidia at 37°C overnight on MAG agar. Female CD1 mice weighing 23 to 25 g (Charles River, Inc., Ltd.) were housed in groups of 5 in cages with access to food and water. The mice were immunosuppressed with two single doses of 1,000 mg $\cdot$ kg$^{-1}$ cortisone acetate (MedChemExpress), which were injected intraperitoneally 3 days before and immediately prior to the infection with conidia (day 0). After being anesthetized with tri-bromoethanol (Aladdin), the immunosuppressed mice were infected via the tracheal instillation of spore suspensions of $1.0 \times 10^5$ conidia in 20 $\mu$L of PBS solution or were injected via the lateral tail vein with 0.1 mL of spore suspensions of $1 \times 10^3$ conidia. As a negative control, a group of 5 mice received PBS tracheal instillation or PBS injection only. The infected animals were weighed every 24 h from the day of infection and were visually inspected twice daily. Mortality was monitored for 14 days.

For the DMG treatment experiments, the infections were done as described, and DMG was administered as a dose of 3 mg dissolved into a 0.2 mL PBS solution. All of the mice were given one oral dose per day. The control group mice were only given sterile PBS. The DMG treatment was performed at 6 h postinfection and then at every 24 h postinfection, up to day 14.

**Histopathology.** For the histological examination, the lungs and livers were excised, fixed in 10% buffered formaldehyde, and then embedded in paraffin. A series of 5 $\mu$m sections were stained using a Grocott's Methenamine Silver (GMS) Stain Kit. The slides were observed, and the images were recorded using a camera (Olympus CK53). To analyze the fungal burden in the tissues, images were selected from 9 GMS-stained lung slices from formalin fixed paraffin-embedded lungs, and then the images of each group of mice were quantified using an image analysis program (ImageJ).

**Statistical analysis.** Statistical differences between the experimental groups were determined via a one-way analysis of variance (one-way ANOVA) that was followed by a Tukey's *post hoc* test. Results involving two experimental groups were analyzed via a Student's *t* test or a Mann-Whitney U test. A Kaplan-Meier survival analysis was used to create a survival curve and to estimate the survival rate over time, and *P* values were calculated through a log-rank analysis (for the comparative survival analysis). A *P* value of <0.05 was considered to be indicative of a statistically significant result. Statistical analyses were performed using the GraphPad Prism 8 software package. (*, $P < 0.05$; **, $P < 0.01$; ***, $P < 0.001$; ****, $P < 0.0001$; ns, not significant).

## SUPPLEMENTAL MATERIAL

Supplemental material is available online only.
**SUPPLEMENTAL FILE 1**, PDF file, 14.4 MB.
**SUPPLEMENTAL FILE 2**, XLSX file, 0.04 MB.

## ACKNOWLEDGMENTS

We thank A.A. Brakhage, T. Heinekamp, and O. Kniemeyer (HKI Jena, Germany) for providing the CEA17D*akuB*$^{KU80}$ strain. We thank Sanhua Fang, Dan Yang, Yanwei Li, Li Liu, Jingyao Chen, and Chengcheng Zhang from the core facility of the Zhejiang University School Of Medicine for their technical support.

This research was supported by the Research Fund for International Young Scientists from the Natural Science Foundation of China under Grant No. 32050410296, the Huadong Medicine Joint Funds of the Zhejiang Provincial Natural Science Foundation of China under Grant No. LHDMZ22H190001, and start-up funds from the School of Medicine and Children's Hospital.

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
