## [Reviewer comments · Microbiology Spectrum]

Microbiology Spectrum

Urease of *Aspergillus fumigatus* is required for survival in macrophages and virulence

Daniel Scharf, Zhenzhen Xiong, Nan Zhang, Liru Xu, Zhiduo Deng, Jarukitt Limwachiranon, Yaojie Guo, Yi Han, and Wei Yang

Corresponding Author(s): Daniel Scharf, Zhejiang University

Review Timeline:

Submission Date:	September 1, 2022
Editorial Decision:	September 20, 2022
Revision Received:	January 16, 2023
Editorial Decision:	January 19, 2023
Revision Received:	February 7, 2023
Accepted:	February 9, 2023

Editor: Gustavo Goldman

Reviewer(s): Disclosure of reviewer identity is with reference to reviewer comments included in decision letter(s). The following individuals involved in review of your submission have agreed to reveal their identity: Norman van Rhijn (Reviewer #1); Jorge Amich (Reviewer #2)

Transaction Report:

DOI: <https://doi.org/10.1128/spectrum.03508-22>

September 20, 2022

Prof. Daniel H Scharf
Zhejiang University
Hangzhou
China

Re: Spectrum03508-22 (Urease of *Aspergillus fumigatus* is required for survival in macrophages and virulence)

Dear Prof. Daniel H Scharf:

Your manuscript was reviewed by two reviewers who suggested many modifications. Please, submit a revised version together with a rebuttal letter addressing point-by-point raised by each reviewer.

Link Not Available

Sincerely,

Gustavo Goldman

Journals Department
Reviewer comments:

Reviewer #1 (Comments for the Author):

In the manuscript Spectrum03508-22 "Urease of *Aspergillus fumigatus* is required for survival in macrophages and virulence" the authors explore urease and the accessory proteins required for urea utilisation. Their mode of action and interaction is described, and the requirement of these product for infection. DMG as a nickel chelator is explored to reduce fungal burden. There is some interesting data in here which can have some wide impact for fungal biology. However, there are some major concerns with interpretation of some of the data and the link between urease and nickel chelation. In addition, several key aspects should be clarified.

There are some concerns with the construction of the reconstituted strains, which should be clarified and more data should be

provided to prove single integration and the differences between wt and reconstituted isolates should be further explored.

The link between DMG and urease is not proven and has likely nothing to do with each other. The data actually supports that nickel chelation and urease essentiality are unlinked. The pro-inflammatory effect of nickel, and therefore chelation causing differences in immunology are interesting but do not relate to the argument of urease being a good antifungal target. More proof is required to link urease and nickel chelation as a therapy.

- L43: most common life-threatening fungal disease. This is not true. Candidiasis is the most common.
- L44 morbidity of invasive aspergillosis continues to increase yearly. Several of these references do not support this statement. Only the US study shows increase in time.
- L51-53 needs a reference.
- L57-59 It is mentioned that eukaryotic ureases are different than bacterial, but then bacterial ureases are expanded upon. Would it not be better to find a well-described eukaryotic system to expand upon in the next paragraph?
- L61 Explain what UreE is, is this the urease? Same goes for HypA, needs a description. Does UreG form a dimer as well? Does UreG then disassociate from UreE to complex with F/D? Needs a better description of the process.
- L72: urea is evenly distributed throughout the human body. This is not true. It is also highly variable between individuals.
- L90-91 Reference 36 is not the primary data. Please cite primary data here.
- L105 ureF needs a AFUA descriptor. The alignment or BLAST result of *A. thaliana* UreF to *A. fumigatus* should be included here.
- L111 "Which are crucial for catalytic activity" this is not proven in *A. fumigatus* so mention in what species this is proven for or remove this statement.
- Figure S1. Species should be italicised. Also the genes/protein IDs per species used for this tree should be included on each line. The result that it supports three distinct and well-supported clades is unfounded. The phylogenetic tree follows the tree of life as is usual with conserved proteins. If the authors want to make the statement about three clades, it needs to be proven statistically.
- Figure S2 States it is the alignment of the whole fungal and plant ureB protein. This is not true, either align the whole protein or mention why only subsection has been shown.
- Similar comment for UreD, why this particular part. How was it selected?
- L115 conserved residues essential for internal nickel transfer. It is not fully conserved because not all species have these residues. Either explain or remove.
- Figure S3 and S4 do not contain full protein, so why these regions and how selected. All these figures need to change.
- Figure S4 and L116 strongest conservation in N-terminal and C-terminal parts. But this is not the N- and C-terminal part of the whole protein?
- L126 CEA17 and CEA17deltaku80 are two different strains. See Bertuzzi et al 2021 and da Silva Ferreira et al 2016, include these refs.
- Figure S7A should include how long the 3' and 5' arms were.
- Figure S7C how long are the UTRs, was this based on the actual transcript length of the genes?
- The PCR strategy of the reconstituted strains does not prove targeted integration into the locus. It also does not prove single copy reintroduction. As differences between Wildtype and reconstituted strains have been found, this needs to be clarified and proven.
- The materials and method show that plasmids were transformed to complement with 1kb upstream and downstream. Promoters can be much longer than 1kb. How was 1kb selected? Any data to back this up? *A. fumigatus* does not retain the pUC plasmid well under non-selective conditions. Integration into the genome is required for these isolates.
- Figure 1C, perform the statics with everything compared to WT, the complemented ureD and ureG isolates will be significantly different. This is problematic.
- L146 "created" prefer to use "constructed".
- The tagged strains should be assessed for growth on urea, expression of their tagged protein to make sure it is functional to the same level as the wildtype.
- L149 up-regulation. This is not measured, more protein is measured.
- L151: eGFP-UreB is localized in the cytoplasm. Higher quality pictures zooming onto one hyphae are required. This doesn't look like localisation but auto-fluorescence to me?
- The tap tagged versions of proteins should be tested if they are functional by phenotypic analysis. The authors say this was done, but results are not shown. It is possible the tag interferes with binding, how was this assessed and why are some N-terminal and others C-terminal?
- The IgG eluate showed enzyme activity? The authors should explain why this is the case and what it means for their results.
- Figure 3A, the ureB has two lanes while all others have one lane? I can see background signal at the top for the ureB. I suspect there is more background cut out from these gels, maybe even for all others. This is manipulation of the data. Show the full gels. A wildtype control should really be included in these experiments.
- Figure 3C; was there absolutely 0 urease activity when using a wt and performed the eluate. There is always some background which is supported by your IgG in the tagged isolates. Was this performed?
- L193 The authors should clarify why a systemic model was used? This is not the normal route of infection and urea might actually be higher in the bloodstream compared to the airspace in the lung?
- L201-202: hyphae had been cleared from the lung. They were infected in the tail vein, so not cleared. Even for wildtype there is

little fungal biomass there.

- The macrophage experiment could represent that the ureB mutant grows slower and therefore spores are more easily killed. Has growth rate been tested in liquid medium?
- L220-223 resting conidia are not metabolically active (therefore resting). Finding a difference in urease is rather odd. Can the authors comment on this further?
- The inhibition of DMG on AMM-nitrate shows that other nickel processes are essential for growth and not just the urease. It proves the essentiality of nickel and not of urease. Figure 7A shows that equal amounts of DMG are required to inhibit growth on both media, unlike what the authors state in L229, which is misrepresenting the data. Therefore, DMG does not impact affect urease and it is an off-target effect.
- Figure S9 no error bars, was this experiment just performed once? Needs to be performed more to prove toxicity of DMG.
- L236: does not affect macrophages. It definitely affects them, but does not affect viability. Processes will be different even with low concentrations of DMG. Nickel induces several pro-inflammatory pathways. It is likely DMG causes a direct effect on macrophages and uptake/killing.
- It is known nickel promotes inflammation through various processes. The reduction of *A. fumigatus* mortality is interesting but unrelated to urease. It likely has nothing to do with the fungus responding but with reducing inflammation, while interesting it does not add to the argument of urease as a drug target.
- L331 nickel chelation therapy. This is overstatement, are there any examples of actual nickel chelation development as therapy?
- L395 PCR product was purified. Gel or column?
- L587 what GMS stain kit was used?

Reviewer #2 (Comments for the Author):

This is a very well conducted study on the role of urease (the enzyme itself and associated proteins) for urea utilization, survival in macrophages and virulence.

I find the study convincing, well executed and relevant.

However, there are a few points that I believe need to be addressed to fully support the conclusions.

MAJOR

1. Section starting in line 153. What other proteins were co-purified with the tagged versions? There seems to be many. Supplementary tables/datafile should be added.

I would argue that a control would be required for this experiment. Did the authors "purify" a mycelium without a tagged protein to check which (if) proteins are detected as background? Even better, an unrelated protein could be tagged to confirm that the Ure proteins of interest do not co-purify.

2. Lines 220-221. I believe this is not very accurate. If the conidia were obtained from a medium containing NO₃ or urea as nitrogen source, they will likely contain urease, because it is expressed in the mycelium (Fig.2A&B) and likely packed. I would expect that is why those conidia protein extracts have urease activity.

Now, the presence of urease inside the conidia would not have the capacity to modulate pH, as conidia are metabolically inactive. Hence, there seem to be two options, either UreB is located at the conidial surface (for which I do not find evidence in doi: 10.1128/mBio.01557-18 or doi: 10.1074/mcp.RA117.000069), or the conidia need to activate metabolism.

This is relevant because, if the conidia were taken from a medium containing NH₄ they would likely not have this activity, due to nitrogen catabolite repression (It would be interesting if the authors could test the expression/protein level in mycelium in the presence of NH₄). Thus, if the protein needs to be in the conidial surface to modulate lysosomal pH, then the origin of the conidia can affect the outcome of the interaction with macrophages. In contrast, if the conidia need to activate metabolism, probably urease is rapidly expressed and the effect would be independent of the origin of conidia.

The authors could try to localize UreB in conidia harvested under different conditions using their GFP strain. Additionally, they could challenge conidia obtained from NH₄ medium and see if they can buffer pH as well.

MINOR

-Lines 60-66. This part is not easy to understand. What are UreE, G and F? Which one is urease? What is HypA? Are these co-regulated? Please, give more details for this section.

-Lines 102-105. The wording here is too informal. Please, modify. Please, add the ID for UreF

-Line 126, The CEA17 isolate is formally the pyrG- Ku80+ strain. Please, clarify which one is the isolate used in the study: ΔakuB KU80 or A1160 pyrG+ (doi: 10.1093/mmy/myaa075)

-Fig S7 and Lines 396-401. As I understand from material and methods, the cassettes to construct the complemented strains harboured the Hyg gene. Was this used as counter-selection versus a previous resistance to pyrithiamine? If this was right, the

text should mention this, and the figure should include the Hyg gene. If not, please, add details about the construction of these strains.

-Fig 1A. Please, indicate the total number of conidia inoculated, not the concentration. As it is not stated what volume is inoculated, the number of conidia is not known.

-Fig 1C. It is common to find that complemented strains do not recover completely wild-type phenotype. However, I find remarkable that ureG and ureD complemented only reconstitute {less than or equal to}50% activity. Did the authors check the sequence of the genes in the cassette?

-Line 173. The results here indicate that UreG do NOT bind to UreB. The authors can only speculate at this point in the manuscript that it may be a transient binding (this is continued in the next section). But the results here do not reflect this.

-Section starting in line 175. It is remarkable that the amount of Ure proteins detected in the absence of UreD and UreG is much smaller. Have the authors quantify this? The authors should mention and discuss this point. May it be that UreD and G are required for protein stability?

-Line 229. I would say that 5mM DMG are required to inhibit growth also on urea.

-Fig. 7. Please, make panel A smaller and enlarge all the others, particularly F. It is quite difficult to see the details.

-Line 285. Other factors could be identified from the full set of proteins co-purified in 3A.

-Line 313. Do the authors discard a role of urease in nutrition during infection? In other words, does the fungus need to acquire nitrogen from urea in the tissues? Are there arguments in favour or against of this? This is quite relevant because the usefulness of nickel chelators would depend on whether urease is only essential in the early stages of infection (i.e. when conidia are being phagocytosed, before hyphae are spread) or during the whole infection process.

Staff Comments:

Preparing Revision Guidelines

Please return the manuscript within 60 days; if you cannot complete the modification within this time period, please contact me. If you do not wish to modify the manuscript and prefer to submit it to another journal, please notify me of your decision immediately so that the manuscript may be formally withdrawn from consideration by Microbiology Spectrum.

Response to the reviewer's comments:

To Reviewer #1

In the manuscript Spectrum03508-22 "Urease of *Aspergillus fumigatus* is required for survival in macrophages and virulence" the authors explore urease and the accessory proteins required for urea utilisation. Their mode of action and interaction is described, and the requirement of these product for infection. DMG as a nickel chelator is explored to reduce fungal burden. There is some interesting data in here which can have some wide impact for fungal biology. However, there are some major concerns with interpretation of some of the data and the link between urease and nickel chelation. In addition, several key aspects should be clarified.

Response: *We are grateful for your precious comments and advice. Those comments are all valuable and very helpful for revising and improving our paper. We have revised the manuscript accordingly, and our point-by-point responses are presented below.*

There are some concerns with the construction of the reconstituted strains, which should be clarified and more data should be provided to prove single integration and the differences between wt and reconstituted isolates should be further explored.

Response: *We performed Southern Blot to show single integration and added the data to the supplements.*

The link between DMG and urease is not proven and has likely nothing to do with each other. The data actually supports that nickel chelation and urease essentiality are unlinked. The pro-inflammatory effect of nickel, and therefore chelation causing differences in immunology are interesting but do not relate to the argument of urease being a good antifungal target. More proof is required to link urease and nickel chelation as a therapy.

Response: *We disagree with this; our additional experiments prove that there is a specific effect of DMG on *A. fumigatus* which is linked to urease activity. We agree that there might be additional effects, especially a more global perturbation of nickel homeostasis.*

1- L43: most common life-threatening fungal disease. This is not true. Candidiasis is the most common.

Response: *Thank you very much for your careful review. We apologize for this error and rewrote the sentence in the revised manuscript as follows: "Invasive aspergillosis (IA), mainly caused by the opportunistic mould *Aspergillus fumigatus*, is one of the most common life-threatening fungal diseases in immunocompromised patients".*

2- L44 morbidity of invasive aspergillosis continues to increase yearly. Several of these references do not support this statement. Only the US study shows increase in time.

Response: *Thank you for pointing out the inaccurate statement. We checked the references and revised the sentence.*

3- L51-53 needs a reference.

Response: We appreciate your suggestion. To be more accurate, we rewrote the sentence.

4- L57-59 It is mentioned that eukaryotic ureases are different than bacterial, but then bacterial ureases are expanded upon. Would it not be better to find a well-described eukaryotic system to expand upon in the next paragraph?

Response: Thank you for your careful review. We agree with the comments on how to expand next paragraph, but the urease systems which are best characterized are all from bacteria. Eukaryotic systems in comparison are much less well characterized so we focus on bacterial systems in the introduction.

For a more logical transition to the next paragraph, we added the contents as follows: "Despite the difference in subunit composition, all ureases possess similar tertiary structures and the urease active site where two nickel ions are coordinated by a carbamylated lysine, four histidine residues and one aspartate residue is highly conserved. Urease is initially synthesized as an inactive apoenzyme, to become enzymatically active, it must undergo a maturation process that involves carbamylation of the active-site lysine followed by delivery of two nickel ions into the active site." at the end of this section, whilst in the next paragraph we provided a brief description for plant urease maturation as well, see next response.

5- L61 Explain what UreE is, is this the urease? Same goes for HypA, needs a description. Does UreG form a dimer as well? Does UreG then disassociates from UreE to complex with F/D? Needs a better description of the process.

Response: To be clearer and in accordance with your concerns, we provided a more detailed description of the maturation process of urease, including what are the roles of the urease accessory proteins UreD, UreE, UreF, and UreG as well as the hydrogenase maturation factor HypA.

As follows:

"The maturation of urease in bacteria is carried out by four accessory proteins named UreD (called UreH in *H. pylori*), UreE, UreF, and UreG. UreE is known to be a dimeric nickel-binding protein that supplies nickel to urease during the maturation process. UreG is a chaperone and a SIMIBI (signal recognition particle, MinD, and BioD) class GTPases, responsible for the GTP hydrolysis associated to the transfer of CO₂ to the active-site lysine. UreF appears to gate the GTPase activity of UreG to enhance the fidelity of urease activation. It has been proposed that nickel-bound UreE dimer can bind two UreG monomers in the presence of GTP and Mg²⁺ to form the UreE₂G₂ complex, triggering nickel translocation from UreE to UreG. UreF forms a UreF₂D₂ complex with UreD in a 2:2 stoichiometry, the UreF₂D₂ complex competes with UreE₂ for nickel-charged UreG₂ to form the UreG₂F₂D₂ complex, which further forms UreG₂F₂D₂-apourease activation complex through a direct interaction between UreD and apourease, then GTP hydrolysis by UreG is catalysed to complete the final step of nickel insertion into apourease. This transfer between the nickel binding site of UreG dimer and the urease is mediated by a transfer tunnel in the UreF₂/D₂ complex and amino acid mutations in the tunnel can greatly reduce the urease activity. In *H. pylori*, UreE can receive its nickel from hydrogenase maturation factor HypA, suggesting a cross-talking exists between the urease and [NiFe]-hydrogenase maturation pathways. Homologues of UreD, UreF and UreG have

also been identified in plants and shown to be essential for urease maturation after a stepwise assembly. Interestingly, a UreE homologue was not found in plants, and the plant UreG with specific HXH motifs was postulated to combine the function of bacterial UreE and UreG." For supporting literatures, please see the bibliography in our manuscript.

6- L72: urea is evenly distributed throughout the human body. This is not true. It is also highly variable between individuals.

Response: We appreciate your careful review and absolutely agree that the urea level is highly variable between individuals, in particular those people with or without renal disease. We have revised the sentence in the manuscript. Literatures supporting the view are listed below:

(Ronne-Engström E et al., 2001 doi: 10.3171/jns.2001.94.3.0397);

(Tyvold SS et al., 2007 doi: 10.1186/1465-9921-8-78)

7- L90-91 Reference 36 is not the primary data. Please cite primary data here.

Response: Thank you for the kind reminding. We have inserted the references containing primary data.

8- L105 ureF needs an AFUA descriptor. The alignment or BLAST result of *A. thaliana* UreF to *A. fumigatus* should be included here.

Response: Since the UreF coding sequence is located between *Afu6g04380* and *Afu6g04390*, we designated it as *Afu6g04385*. Following your suggestion, we added the alignment result as **figure S1**. And we also rewrote the last sentence of this part as follows:

"Nevertheless, we identified an ortholog of *ureF* in *A. fumigatus* genome via the tblastn program using *Arabidopsis thaliana* UreF as a query (30% amino acid identity, 50% amino acid similarity; **figure S1**) and designated this new gene as *Afu6g04385*."

9- L111 "Which are crucial for catalytic activity" this is not proven in *A. fumigatus* so mention in what species this is proven for or remove this statement.

Response: We are grateful for the above suggestion. This statement has been revised in the manuscript as follows: "which are crucial for the catalytic activity of bacterial and plant ureases".

10- Figure S1. Species should be italicised. Also, the genes/protein IDs per species used for this tree should be included on each line. The result that it supports three distinct and well-supported clades is unfounded. The phylogenetic tree follows the tree of life as is usual with conserved proteins. If the authors want to make the statement about three clades, it needs to be proven statistically.

Response: Thanks for your kind reminder. We have corrected the format of the species and added the genes and protein IDs from UniProt. It is true that the three clades follow the tree of life. However, there are still three distinct clades supported by the bootstrap values. The major point is that fungal ureases are distinct from bacterial or plant ureases which might be relevant for the development of specific inhibitors.

11- **Figure S2** States it is the alignment of the whole fungal and plant ureB protein. This is not true, either align the whole protein or mention why only subsection has been shown.

Response: We appreciate your comment. The fact is that we have done the alignment with the whole fungal and plant urease proteins. Considering the large space for displaying the complete alignment result, we decided to show the conserved part of interest, which is also common in other published research papers. In accordance with your concern, we provided the original alignment result as a PDF file (“UreB alignment”) for your reference.

12- Similar comment for UreD, why this particular part. How was it selected?

Response: We only presented the subsection containing the conserved residues that had been genetically and biochemically confirmed to be essential for urease activation in bacteria. These residues were proposed to support an internal nickel-transfer tunnel.

(Farrugia MA et al., 2015 doi: 10.1021/acs.biochem.5b00942)

(Musiani F et al., 2017 doi: 10.1021/acs.jctc.7b00042)

(Masetti M et al., 2021 doi: 10.1016/j.jinorgbio.2021.111554)

Please find the original alignment data in the attached PDF file “UreD alignment” if you are interested.

13- L115 conserved residues essential for internal nickel transfer. It is not fully conserved because not all species have these residues. Either explain or remove.

Response: We are sorry for our incorrect statement about the conserved residues in UreD. It has been revised as follows:

“A. fumigatus UreD shares a very low level of sequence similarity with its homologs (figure S4), but has conserved residues that have been genetically and biochemically confirmed to be essential for urease activation in bacteria and proposed to be involved in internal nickel transfer”

Explanations to the comment “not fully conserved because not all species have these residues” are listed below:

Among the second marked “Ts”, two distinct residues are the “E83” in *HpUreD* and the “S85” in *KaUreD*. According to Farrugia MA et al., 2015 (doi: 10.1021/acs.biochem.5b00942) the “E83” and “S85” are positioned at the termini of the predicted tunnels in *HpUreD* and *KaUreD*. They were identified as crucial residues for urease activation.

Threonine and Serine belong to polar amino acids and have similar sidechains, and therefore should have the same biochemical functions in proteins.

The distinct residue out of the last two marked conserved “Ds” is “E”. Both Aspartic acid and Glutamic acid have negatively charged sidechains, also supposed to function equally.

The last debate is about the “Qs”. A conservation analysis on Arg95 of *HpUreD* showed that “Arg95 of is present in only 1% of the cases, while in 96% of the sequences it is mutated with a glutamine, which is nevertheless able to form a H-bond with the residue in the position of Glu140 through its –NH₂ group.” (Musiani F et al., 2017 doi: 10.1021/acs.jctc.7b00042)

In conclusion, these conservative mutations do not alter the function of UreD protein.

14- Figure S3 and S4 do not contain full protein, so why these regions and how selected. All

these figures need to change.

Response: Similarly, we selectively displayed conserved regions of interest. For reference, complete alignment files “UreE alignment”, “UreF alignment”, “UreG alignment” are attached this time.

15- Figure S4 and L116 strongest conservation in N-terminal and C-terminal parts. But this is not the N- and C-terminal part of the whole protein?

Response: Thank you for the question. We corrected this statement as: “Sequence comparison of UreF homologs revealed the strongest conservation in the C-terminal part **and the region near the N terminus** (figure S5), which encompass the residues constituting the UreG binding site.” Please see the file “UreF alignment” for the full alignment data.

16- L126 CEA17 and CEA17deltaku80 are two different strains. See Bertuzzi et al 2021 and da Silva Ferreira et al 2016, include these refs.

Response: We thank you for pointing this out and apologize for our carelessness. We used the strain CEA17 Δ akuB^{KU80} as the wild type in this study and corrected the wrong strain information in our manuscript, also the references were added.

17- Figure S7A should include how long the 3' and 5' arms were.

Response: We gratefully thank for the kind advice. The size of the two arms is approximately 1 kb, and has been added in the related figures. We determined the arm length following a previous study by Krappmann S, et al. (doi: 10.1128/EC.5.1.212-215.2006).

18- Figure S7C how long are the UTRs, was this based on the actual transcript length of the genes?

Response: Actually, we mislabeled the flanks we choose for recombination as UTR. These sequences are actually the intergenic regions, based on the distance to the next annotated gene and not on transcript length. We listed the flank lengths of *ureB*, *ureD*, *ureF* and *ureG* in the following table. When we constructed the pUC-gene cassette plasmids for the complementation of Δ *ureB*, Δ *ureD* and Δ *ureG*, the region covering complete 5' flank sequence -3' flank of each gene was cloned into the pUC-hph vector. However, for the construction of Δ *ureF*+*ureF*, gene *ureF* with a *Hyg*^R cassette at the stop codon was reintroduced via homologous recombination based on \approx 1kb of the 3' and 5' arms.

Gene	AFU No.	Flank length (bp)	
		5' flank	3' flank
ureB	Afu1g04560	794	188
ureD	Afu2g16070	1037	512
ureF	Afu6g04385	326	759
ureG	Afu2g12900	1195	258

19- The PCR strategy of the reconstituted strains does not prove targeted integration into the

locus. It also does not prove single copy reintroduction. As differences between Wildtype and reconstituted strains have been found, this needs to be clarified and proven.

Response: The above comments are valuable and remind us to further verify the complemented strains. Therefore, we performed Southern blot analysis. The results showed the ectopic integration of pUChph-*ureB*, pUChph-*ureD* and pUChph-*ureG*. The reconstituted isolate $\Delta ureF+ureF$ constructed by homologous recombination with the native *ureF* gene fused with a Hyg^R cassette was also verified by Southern blot. Details, please refer to figure S8-S11.

20- The materials and method show that plasmids were transformed to complement with 1kb upstream and downstream. Promoters can be much longer than 1kb. How was 1kb selected? Any data to back this up? *A. fumigatus* does not retain the pUC plasmid well under non-selective conditions. Integration into the genome is required for these isolates.

Response: Thank you so much for your review. It is really true as you mentioned that promoters can be much longer than 1kb. In fact, for complementation of mutants $\Delta ureB$, $\Delta ureD$, and $\Delta ureG$, we followed the method by Wartenberg, D et al., 2011 (doi: 10.1016/j.ijmm.2011.04.016). It was the region containing full 5' intergenic region + coding gene + 3' intergenic region (not 1kb upstream and downstream) cloned into pUChph plasmid, which was used for reintroducing each gene into the corresponding mutant. Hence, for genes *ureB*, *ureD* and *ureG*, their complete promoters and terminators are included in the plasmids for transformation. And the $\Delta ureF$ strain was complemented by homologous recombination with the native *ureF* gene fused with a Hyg^R cassette, replacing the *ptrA* cassette. The mistakes in the manuscript have been corrected.

As for how the pUC plasmids are kept in cells, we agree with your comment. The bacterial pUC plasmids contains pMB1 replication origin, which is unable to reproduce in eukaryotic cells. In the case of *A. fumigatus*, 10-20 μ g pUC plasmids were used, only dozens of transformants could be obtained at one time. This suggests that the plasmids were incorporated into host genome through ectopic integration to be hygromycin resistant.

21- Figure 1C, perform the statics with everything compared to WT, the complemented *ureD* and *ureG* isolates will be significantly different. This is problematic.

Response: We are very grateful to your comment. With regard to the significant difference in urease activities between wt and the reconstituted strains $\Delta ureD+ureD$ and $\Delta ureG+ureG$, we assumed that this is because of a random integration of the pUC plasmids into the inactive region on *A. fumigatus* genome, where the gene expression is decreased. Another possibility is that the promotor was shorten during single-crossover recombination, which could lower the transcription efficiency. Thus, we recovered the other two transformants of each strain from glycerol stocks and measured the enzyme activities. Finally, we identified strains with urease activity comparable to wild type (as shown below). In the revised version, we substituted the results of $\Delta ureD+ureD$ #5 and $\Delta ureG+ureG$ #3 for previous results from transformants $\Delta ureD+ureD$ #4 and $\Delta ureG+ureG$ #1.

22- L146 "created" prefer to use "constructed".

Response: Thanks for your nice suggestion. We have replaced "created" with "constructed".

23- The tagged strains should be assessed for growth on urea, expression of their tagged protein to make sure it is functional to the same level as the wildtype.

Response: Thank you for the comment. We performed growth and enzyme assay for *EGFP::ureB* strain. As shown below, *EGFP::ureB* is functional to the same level as the wildtype. More details please refer to figure S13.

24- L149 up-regulation. This is not measured, more protein is measured.

Response: Thanks for the kind reminding. This has been revised in the new manuscript as follows: "As a consequence of the increase in *UreB* expression, urease activity of the strain cultured in AMM urea was significantly higher than that in AMM nitrate".

25- L151: eGFP-UreB is localized in the cytoplasm. Higher quality pictures zooming onto one hyphae are required. This doesn't look like localisation but auto-fluorescence to me?

Response: We gratefully appreciate for your professional suggestion. As suggested, we have improved the quality of the pictures and replaced the old ones in the manuscript.

26- The tap tagged versions of proteins should be tested if they are functional by phenotypic analysis. The authors say this was done, but results are not shown. It is possible the tag interferes with binding, how was this assessed and why are some N-terminal and others C-

terminal?

Response:

We are very sorry for our negligence of showing the phenotypic analysis results. In the newly submitted manuscript we have added the related data in figure S15.

We totally understand your concern that the TAP tag would interfere with the interaction between apo urease and the accessory proteins. This is exactly why we determined the TAP-tag site of the proteins based on the alignment results. Considering the potential function of the conserved terminal regions, we preferentially put the TAP tag at the less- or non-conserved ends for all the proteins.

Like plant and other fungal ureases, *A. fumigatus* UreB is a long peptide (838 bp) forming the functional monomer of apo urease. The following schematic comparison shows that amino-terminal residues of the monomers of plant and fungal enzymes are similar to the small subunits of bacterial enzymes, while the large subunits of bacterial ureases resemble the carboxy-terminal portions of plant and fungal subunits (Sirko, A and Brodzik, R, 2000 doi: 10.18388/abp.2000_3972). It was proved previously that the catalytic site is located in the large subunit of bacterial enzymes and in the respective regions of ureases from eukaryotic organisms, which was also showed in our alignment result of UreB (figure S3 in revised version). Therefore, we fused the TAP tag at the N terminus of UreB. Same consideration for the conserved functional end applies to UreD, UreF and UreG.

Growth phenotype on urea agar and enzyme activity assay showed no significant differences between wild type and the two *nTap*-tagged strains and *ureD::cTap* strain, whereas *ureG::cTap* showed slower growth rate and decreased urease activity. The Tap tag (~20 KD) probably influences the binding of UreG to other accessory proteins to some extent due to space steric hindrance, which might be minimized by using small tag (e.g., Strep-Tag II). However, the phenotypic analysis data of *ureG::cTap* would not challenge our conclusion on the transient interaction of UreG with apo urease and UreD/F because no UreG was found in

the purification experiment of the other three Tap-tagged strains.

Figure 1. Schematic comparison of the structural subunits of ureases from selected organisms.

27- The IgG eluate showed enzyme activity? The authors should explain why this is the case and what it means for their results.

Response: We apologize for not describing this part clearly. Here is a more detailed explanation:

The tandem affinity purification (TAP) method is based on a recombinant fusion tag composed of two protein A domains and a calmodulin binding peptide (CBP) separated by a tobacco etch virus protease (TEV) cleavage site. As stated in Materials and Methods, the TAP tag purification underwent two steps of purification with IgG sepharose and calmodulin affinity resin, resulting in IgG eluate (≈ 2 mL) and CaM eluate (≈ 1 mL).

We tested if the eluted fractions from both purification steps have ureolytic activity and showed that both the IgG eluate and the CaM eluate of nTAP-UreB, UreD-cTAP or nTAP-UreF demonstrated marked enzyme activities at different levels, but the UreG-cTAP enriched fraction did not present any activity.

As shown, the detected enzyme activity of IgG eluate is less than that of CaM eluate (1/2 in *nTap::ureB* group), which is consistent with their protein concentrations.

28- Figure 3A, the ureB has two lanes while all others have one lane? I can see background signal at the top for the ureB. I suspect there is more background cut out from these gels, maybe even for all others. This is manipulation of the data. Show the full gels. A wildtype control should really be included in these experiments.

Response: We really appreciate your careful review and precious comments. In order to resolve your concerns, we show the original pictures of silver staining and Western Blot below.

Fig 3A-top. The IgG/CaM eluates from the indicated *A. fumigatus* strains were analyzed by 4%-12% PAGE with silver staining. UreB/D/F/G are marked on the gels.

Fig 3A-bottom. The Tap-tagged proteins in IgG/CaM eluates were detected by Western Blot probed with the anti-CBP antibody. The arrowheads on the former three membranes indicate the MBP-TAP control. The lane of TAP control on the last membrane was removed due to overexposure.

Answers to the above questions are listed as follows:

1. After TAP tag purification, we analyzed the IgG and CaM eluates by SDS-PAGE with silver staining and Western Blot. Only CaM eluates showed discernible bands of target proteins. Due to low concentration, the TAP-tagged proteins except for UreB* in IgG eluates were not detectable by Western Blot. Whereas, before applied to detection all the CaM eluates were precipitated by using trichloroacetic acid (TCA) and dissolved in a small volume of buffer (50-100 μ L) to get concentrated proteins. Therefore, all proteins could be detected.
2. Indeed, all elutes on silver-stained gels have background, which is inevitable because of

the high sensitivity of silver staining. We added the MS datafile including detailed protein information of CaM eluates from wild type and the four TAP-tagged strains in the newly submitted supplementary materials for reference. As we expected, none of the Ure proteins of interest was found in the eluate from wild type.

3. We had a wild type control in the experiment but didn't show the silver staining result. We are very sorry for our mistake and added it in the revised manuscript.

Because showing the results concerning IgG eluates does not contribute very much to our study, but rather confuses the readers, we decided to remove them in the revised manuscript. We really hope that you can take our explanation into consideration and agree to our final decision.

29- Figure 3C; was there absolutely 0 urease activity when using a wt and performed the eluate. There is always some background which is supported by your IgG in the tagged isolates. Was this performed?

Response: The activity of the wt and UreG-TAP were not detectable. As you can see the activity of the other eluates is also very low in comparison to purified TAP-UreB.

30- L193 The authors should clarify why a systemic model was used? This is not the normal route of infection and urea might actually be higher in the bloodstream compared to the airspace in the lung?

Response: In accordance with other publications we used a systemic model which is well established for the identification of *A. fumigatus* virulence factors. However, for the revised manuscript we established a lung infection model and included the results.

31- L201-202: hyphae had been cleared from the lung. They were infected in the tail vein, so not cleared. Even for wildtype there is little fungal biomass there.

Response: We used "almost" which indicates that there is still evidence of fungal biomass. We show an analysis of the fungal burden based on the stained hyphae to compare wild type and deletion mutant.

32- The macrophage experiment could represent that the ureB mutant grows slower and therefore spores are more easily killed. Has growth rate been tested in liquid medium?

Response: Thanks very much for your comment. According to our growth rate data on AMM agar and in AMM liquid medium (with nitrate as nitrogen source), no significant difference was found between wild type and $\Delta ureB$ strain.

33- L220-223 resting conidia are not metabolically active (therefore resting). Finding a difference in urease is rather odd. Can the authors comment on this further?

Response: Thank you very much for this comment. We do not claim that conidia are metabolically active as they represent a dormant stadium. However, they cannot be completely inactive because they need to perceive and integrate signals from the

environment and must be able to start germination under the right circumstances. We could show that the EGFP-UreB fusion shows a clear signal in conidia in comparison to the wild type which is additional prove for the presence of urease in conidia.

34- The inhibition of DMG on AMM-nitrate shows that other nickel processes are essential for growth and not just the urease. It proves the essentiality of nickel and not of urease. Figure 7A shows that equal amounts of DMG are required to inhibit growth on both media, unlike what the authors state in L229, which is misrepresenting the data. Therefore, DMG does not impact affect urease and it is an off-target effect.

Response: We included the colony diameter on both media with DMG which clearly shows that the inhibition is stronger on urea medium. This effect is due to the inhibition of urease. We agree that there is also inhibition on nitrate plates which shows that nickel might be important for other functions. However, the importance of urease and nickel are not mutual exclusive they rather represent different effects.

35- Figure S9 no error bars, was this experiment just performed once? Needs to be performed more to prove toxicity of DMG.

Response: The experiment had been done in triplicate. In the revised manuscript, we have added error bars in the figure.

36- L236: does not affect macrophages. It definitely affects them, but does not affect viability. Processes will be different even with low concentrations of DMG. Nickel induces several pro-inflammatory pathways. It is likely DMG causes a direct effect on macrophages and uptake/killing.

Response: We found that DMG cannot increase the uptake/killing capacity of macrophages. We did killing/phagocytosis assays with *E. coli*, *C. albicans* and SiO₂ microspheres after DMG treatment and found no significant difference. So, we conclude that the DMG effect is specific for *A. fumigatus*.

37- It is known nickel promotes inflammation through various processes. The reduction of *A. fumigatus* mortality is interesting but unrelated to urease. It likely has nothing to do with the fungus responding but with reducing inflammation, while interesting it does not add to the argument of urease as a drug target.

Response: The histopathology shows that after DMG treatment the fungal burden is much lower, also urease deletion has a clear effect on virulence. According to this urease could be a potential drug target.

38- L331 nickel chelation therapy. This is overstatement, are there any examples of actual nickel chelation development as therapy?

Response: We changed our manuscript according to the comment. According to our knowledge there are already FDA approved urease inhibitors and their use for the treatment of infections has been suggested.

39- L395 PCR product was purified. Gel or column?

Response: We apologize for the omission of the details. All the PCR products for further experiments were separated by agarose gel followed by DNA purification using a PCR Clean-Up Kit (Axygen) with columns.

40- L587 what GMS stain kit was used?

Response: The GMS stain kit was prepared by the staff of the School of Medicine core facility.

Response to the reviewer's comments:

To Reviewer #2:

This is a very well conducted study on the role of urease (the enzyme itself and associated proteins) for urea utilization, survival in macrophages and virulence.

I find the study convincing, well executed and relevant.

However, there are a few points that I believe need to be addressed to fully support the conclusions.

Response: We are grateful for your positive comments and advice. Those comments are all valuable and very helpful for revising and improving our paper. We have revised the manuscript accordingly, and our point-by-point responses are presented below.

MAJOR

1. Section starting in line 153. What other proteins were co-purified with the tagged versions? There seems to be many. Supplementary tables/datafile should be added.

I would argue that a control would be required for this experiment. Did the authors "purify" a mycelium without a tagged protein to check which (if) proteins are detected as background? Even better, an unrelated protein could be tagged to confirm that the Ure proteins of interest do not co-purify.

Response: We included a table with all identified proteins from wild type and the differently tagged strains as supplementary material. Furthermore, we checked the purification of other TAP-tagged proteins and did not find urease or maturation protein (these data are not included because they are part of another publication).

2. Lines 220-221. I believe this is not very accurate. If the conidia were obtained from a medium containing NO₃ or urea as nitrogen source, they will likely contain urease, because it is expressed in the mycelium (Fig.2A&B) and likely packed. I would expect that is why those conidia protein extracts have urease activity.

Now, the presence of urease inside the conidia would not have the capacity to modulate pH, as conidia are metabolically inactive. Hence, there seem to be two options, either UreB is located at the conidial surface (for which I do not find evidence in doi: 10.1128/mBio.01557-18 or doi: 10.1074/mcp.RA117.000069), or the conidia need to activate metabolism.

This is relevant because, if the conidia were taken from a medium containing NH₄ they would likely not have this activity, due to nitrogen catabolite repression (It would be interesting if the authors could test the expression/protein level in mycelium in the presence of NH₄). Thus, if the protein needs to be in the conidial surface to modulate lysosomal pH, then the origin of the conidia can affect the outcome of the interaction with macrophages. In contrast, if the conidia need to activate metabolism, probably urease is rapidly expressed

and the effect would be independent of the origin of conidia.

The authors could try to localize UreB in conidia harvested under different conditions using their GFP strain. Additionally, they could challenge conidia obtained from NH₄ medium and see if they can buffer pH as well.

Response: Thank you very much for this comment. We do not claim that conidia are metabolically active as they represent a dormant stadium. However, they cannot be completely inactive because they need to perceive and integrate signals from the environment and must be able to start germination under the right circumstances. We could show that the EGFP-UreB fusion shows a clear signal in conidia in comparison to the wild type which is additional prove for the presence of urease in conidia. As suggested we also compared conidia from different nitrogen sources based on their ability to survive in macrophages and could measure a significant difference between conidia from urea or ammonium plates. In summary, we agree that the urease could be produced during conidiation and is packed into the spores. However, whether the enzyme is somehow located on the surface or secreted in order to affect the phagosome pH cannot be determined here.

MINOR

-Lines 60-66. This part is not easy to understand. What are UreE, G and F? Which one is urease? What is HypA? Are these co-regulated? Please, give more details for this section.

Response: To be clearer and in accordance with your concerns, we provided a more detailed description of the maturation process of urease, including what are the roles of the urease accessory proteins UreE, UreF, and UreG as well as the hydrogenase maturation factor HypA. As follows:

“The maturation of urease in bacteria is carried out by four accessory proteins named UreD (called UreH in *H. pylori*), UreE, UreF, and UreG. UreE is known to be a dimeric nickel-binding protein that supplies nickel to urease during the maturation process. UreG is a chaperone and a SIMIBI (signal recognition particle, MinD, and BioD) class GTPases, responsible for the GTP hydrolysis associated to the transfer of CO₂ to the active-site lysine. UreF appears to gate the GTPase activity of UreG to enhance the fidelity of urease activation. It has been proposed that nickel-bound UreE dimer can bind two UreG monomers in the presence of GTP and Mg²⁺ to form the UreE₂G₂ complex, triggering nickel translocation from UreE to UreG. UreF forms a UreF₂D₂ complex with UreD in a 2:2 stoichiometry, the UreF₂D₂ complex competes with UreE₂ for nickel-charged UreG₂ to form the UreG₂F₂D₂ complex, which further forms UreG₂F₂D₂-apourease activation complex through a direct interaction between UreD and apourease, then GTP hydrolysis by UreG is catalysed to complete the final step of nickel insertion into apourease. This transfer between the nickel binding site of UreG dimer and the urease is mediated by a transfer tunnel in the UreF₂/D₂ complex and amino acid mutations in the tunnel can greatly reduce the urease activity. In *H. pylori*, UreE can receive its nickel from hydrogenase maturation factor HypA, suggesting a cross-talking exists between the urease and [NiFe]-hydrogenase maturation pathways. Homologues of UreD, UreF and UreG have also been identified in plants and shown to be essential for urease maturation after a stepwise assembly. Interestingly, a UreE homologue was not found in plants, and the plant UreG with specific HXH motifs was postulated to combine the function of bacterial UreE and

UreG.” For supporting literatures, please see the bibliography in our manuscript.

-Lines 102-105. The wording here is too informal. Please, modify. Please, add the ID for UreF

Response: We gratefully appreciate for your suggestion. Since the UreF coding sequence is located between *Afu6g04380* and *Afu6g04390*, we designated it as *Afu6g04385*. Also, we modified the sentence in the manuscript as follows:

“Nevertheless, we identified an ortholog of *ureF* in the *A. fumigatus* genome via the tblastn program using *Arabidopsis thaliana* UreF as a query (30% amino acid identity, 50% amino acid similarity; figure S1) and designated this new gene as *Afu6g04385*.”

-Line 126, The CEA17 isolate is formally the pyrG- Ku80+ strain. Please, clarify which one is the isolate used in the study: Δ akuB KU80 or A1160 pyrG+ (doi: 10.1093/mmy/myaa075)

Response: We thank you for pointing this out and apologize for our carelessness. We used the strain CEA17 Δ akuB^{KU80} as the wild type in this study and corrected the wrong strain information in our manuscript, also the reference you mentioned was added.

-Fig S7 and Lines 396-401. As I understand from material and methods, the cassettes to construct the complemented strains harboured the Hyg gene. Was this used as counter-selection versus a previous resistance to pyrithiamine? If this was right, the text should mention this, and the figure should include the Hyg gene. If not, please, add details about the construction of these strains.

Response: Thanks for your thoughtful comments, and our reply is as follows: We constructed the complemented strains Δ ureB+ureB, Δ ureD+ureD, and Δ ureG+ureG following a previous publication by Wartenberg, D et al. (2011) (doi: 10.1016/j.ijmm.2011.04.016). Gene expression cassettes (complete 5' intergenic- coding sequence -3' intergenic of *ureB*, *ureD* and *ureG*) were cloned into pUChph at the *Xba*I or *Hind*III sites (upstream of Hyg^R), yielding pUChph-gene plasmids (See pUChph-ureB below as example). Then, transformation of the mutants with corresponding plasmids resulted in the complemented strains selected by hygromycin resistance (As the pUC plasmids with pMB1 replication origin are unable to reproduce in *A. fumigatus* cells, it needs to be incorporated into host genome through ectopic integration to response hygromycin stress) and urea utilization (to confirm the reconstitution of urease activity). To be clearer, we cited the above reference in our revised manuscript. And it needs to be clarified that the Δ ureF strain was complemented by homologous recombination with the native *ureF* gene fused with a Hyg^R cassette, replacing the ptrA cassette.

The inaccurate information in Materials and Methods with respect to the construction of complemented strains has been corrected, more details also added.

-Fig 1A. Please, indicate the total number of conidia inoculated, not the concentration. As it is not stated what volume is inoculated, the number of conidia is not known.

Response: Thank you for the comment, we have indicated the total number of conidia on each plate in all the related data.

-Fig 1C. It is common to find that complemented strains do not recover completely wild-type phenotype. However, I find remarkable that ureG and ureD complemented only reconstitute {less than or equal to}50% activity. Did the authors check the sequence of the genes in the cassette?

Response: Thank you for your question. With regard to the significant difference in urease activities between wt and the reconstituted strains $\Delta ureD+ureD$ and $\Delta ureG+ureG$, we assumed that this is because of a random integration of the pUC plasmids into the inactive region on *A. fumigatus* genome, where the gene expression might be insufficient. Another possibility is that the promoter was shortened during single-crossover recombination, which could lower the transcription efficiency. Thus, we recovered the other two transformants of each strain from glycerol stocks and measured the enzyme activities. Finally, we identified strains with urease activity comparable to wild type (as shown below). In the revised version, we substituted the results of $\Delta ureD+ureD$ #5 and $\Delta ureG+ureG$ #3 for previous results from transformants $\Delta ureD+ureD$ #4 and $\Delta ureG+ureG$ #1.

Furthermore, we performed Southern blot analysis to further verify the complemented strains. The results showed the ectopic integration of pUChph-ureB, pUChph-ureD and pUChph-ureG. The reconstituted isolate $\Delta ureF+ureF$ constructed by homologous recombination with the native *ureF* gene fused with a Hyg^R cassette was also verified by Southern blot. Details, please refer to figure S8-S11.

-Line 173. The results here indicate that UreG do NOT bind to UreB. The authors can only speculate at this point in the manuscript that it may be a transient binding (this is continued in the next section). But the results here do not reflect this.

Response: From our observations we conclude that UreG is necessary for UreB maturation but it could not be co-purified. Therefore, we conclude that UreG interacts at a certain stage with other maturation proteins but not necessarily with UreB. However, UreG also could not be co-purified with UreF or UreD so the interaction must be weak or very transient and it is known that such interactions might be too weak for TAP co-purification.

-Section starting in line 175. It is remarkable that the amount of Ure proteins detected in the absence of UreD and UreG is much smaller. Have the authors quantify this? The authors should mention and discuss this point. May it be that UreD and G are required for protein stability?

Response: Thanks very much for your careful review. We have to clarify that the above issue resulted from the different protein loading amounts. We loaded the whole amount of purified protein on the SDS-gel. We agree that the UreD and UreG might increase the amount of active urease through stabilization. We did not follow up on this because the major conclusion is that all maturation proteins are required for enzymes activity. We will follow up on this observation in our future research.

-Line 229. I would say that 5mM DMG are required to inhibit growth also on urea.

Response: We measured the colony diameter on urea DMG plates and agree that 5 mM is needed for significant growth reduction. We changed the manuscript accordingly.

-Fig. 7. Please, make panel A smaller and enlarge all the others, particularly F. It is quite difficult to see the details.

Response: Thank you for the suggestion. We have already readjusted the figures according to your suggestion.

-Line 285. Other factors could be identified from the full set of proteins co-purified in 3A.

Response: It is correct that many other background proteins were identified along with the Ure proteins. We have attached the datafile of full protein information as supplementary material for reference.

-Line 313. Do the authors discard a role of urease in nutrition during infection? In other words, does the fungus need to acquire nitrogen from urea in the tissues? Are there arguments in favour or against of this? This is quite relevant because the usefulness of nickel chelators would depend on whether urease is only essential in the early stages of infection (i.e., when conidia are being phagocytosed, before hyphae are spread) or during the whole infection process.

Response: Thank you for your comment, we considered the effect on growth and we hypothesize that urease function is also important in later stages of infection. We assume that urea from the host could be utilized as a substrate and also the use of host amino acids might lead to urea production which needs to be degraded by urease. In addition, the ammonia produced by the urease was shown to be toxic for many cell types and could contribute to host cell damage and invasive growth.

-

January 19, 2023

Prof. Daniel H Scharf
Zhejiang University
Hangzhou
China

Re: Spectrum03508-22R1 (Urease of *Aspergillus fumigatus* is required for survival in macrophages and virulence)

Dear Prof. Daniel H Scharf:

There are still some suggestions of the reviewer #2. Please, address them and submit your revised version together with a rebuttal letter addressing point by point raised by reviewer #2.

Link Not Available

Sincerely,

Gustavo Goldman

Journals Department
Reviewer comments:

Reviewer #1 (Comments for the Author):

I would like to commend the authors on the additional experiments performed and changes made, significantly improving the manuscript and answering the questions raised by this reviewer.

Reviewer #2 (Comments for the Author):

The authors have made a significant effort to respond to my comments. The article has improved significantly and I am almost completely satisfied. However, there are still a couple of minor issues that I would like to ask the authors to address to round off

the manuscript.

1) The origin of urease for its role in macrophage killing of conidia is still not clear.

The authors have measured the macrophage killing capacity on spores grown on different N-sources, and found that conidia grown on ammonium are killed significantly more. However, they report levels of killing of around 80% in fig S17, but of around 25% in 6B, how is this possible?

Besides, the authors should also show if the levels of GFP-UreB in the conidia are lower when the fungus was grown on NH₄, compared to urea. This would imply that the pre-existing urease in conidia (not newly synthesised) is important for the survival in macrophages. Would this imply that conidia grown on urease are more fit for infection?

2) With respect to the co-purification experiments, please, include the reference to the table of purified proteins in the text. The use of another tagged protein is a required control. If the data belongs to a different paper, please, just say what protein was tagged and specify that it did not co-purify the Ure proteins.

3) Given the strong effect of DMG on *A. fumigatus* growing on nitrate, the authors should clearly mention that the nickel chelating strategy seems to be very relevant on other Ni-dependent enzymes. Can the authors discuss which proteins could be affected? Given the relatively small number of Ni-dependent proteins, could the authors compensate the potential effect of DMG on other proteins (for instance, with supplements, or forcing fermentation)

Staff Comments:

Preparing Revision Guidelines

Please return the manuscript within 60 days; if you cannot complete the modification within this time period, please contact me. If you do not wish to modify the manuscript and prefer to submit it to another journal, please notify me of your decision immediately so that the manuscript may be formally withdrawn from consideration by Microbiology Spectrum.

1) The origin of urease for its role in macrophage killing of conidia is still not clear.

The authors have measured the macrophage killing capacity on spores grown on different N-sources, and found that conidia grown on ammonium are killed significantly more. However, they report levels of killing of around 80% in fig S17, but of around 25% in 6B, how is this possible?

Besides, the authors should also show if the levels of GFP-UreB in the conidia are lower when the fungus was grown on NH₄, compared to urea. This would imply that the pre-existing urease in conidia (not newly synthesised) is important for the survival in macrophages. Would this imply that conidia grown on urease are more fit for infection?

> The difference in killing between the figures can be explained by the origin of the conidia from different growth media. If the conidia are derived from a complete medium (here malt extract agar) they are more resistant to killing than conidia from minimal medium which was already shown before in other publications. (for reference please see <https://www.frontiersin.org/articles/10.3389/fmicb.2019.00854/full> and <https://journals.plos.org/plosgenetics/article?id=10.1371/journal.pgen.1008551>)

We included more pictures in the supplement to show that the amount of GFP-UreB in conidia grown on AMM ammonium is very low. We conclude from our results that conidia which contain urease are more resistant to killing and should have an advantage during infection.

2) With respect to the co-purification experiments, please, include the reference to the table of purified proteins in the text.

The use of another tagged protein is a required control. If the data belongs to a different paper, please, just say what protein was tagged and specify that it did not co-purify the Ure proteins.

>To clarify this, we included a reference to the MS table in the respective part of the manuscript and mentioned the comparison with a TAP-tagged version of the NOT complex which does not show co-purification of urease or associated proteins.

3) Given the strong effect of DMG on *A. fumigatus* growing on nitrate, the authors should clearly mention that the nickel chelating strategy seems to be very relevant on other Ni-dependent enzymes. Can the authors discuss which proteins could be affected? Given the relatively small number of Ni-dependent proteins, could the authors compensate the potential effect of DMG on other proteins (for instance, with supplements, or forcing fermentation)

>Thank you for this comment. We mentioned it now in the discussion and gave a short list of potential candidate proteins. It might be possible to identify other nickel-binding proteins using the suggested methods and we will consider this comment in the design of further studies.

February 9, 2023

Prof. Daniel H Scharf
Zhejiang University
Hangzhou
China

Re: Spectrum03508-22R2 (Urease of *Aspergillus fumigatus* is required for survival in macrophages and virulence)

Dear Prof. Daniel H Scharf:

The manuscript is ready for publication. Congratulations !!!!!

Your manuscript has been accepted, and I am forwarding it to the ASM Journals Department for publication. You will be notified when your proofs are ready to be viewed.

Sincerely,

Gustavo Goldman
Editor, Microbiology Spectrum
